

# Active and inactive Andean rock glacier geophysical signatures by comparing 2D joint inversion routines of electrical resistivity and refraction seismic tomography

Giulia de Pasquale[1], Rémi Valois[2], Nicole Schaffer[1], and Shelley MacDonell[1]

[1]Centro de Estudios Avanzados en Zonas Áridas – CEAZA, Raúl Bitrán 1305, La Serena, Chile
[2]Environnement Méditerranéen et Modélisation des Agro-Hydrosystèmes, Université de Avignon - EMMAH, Domaine Saint-Paul, Site Agroparc 228, Avignon, France

**Correspondence:** Giulia de Pasquale (giulia.depasquale@ceaza.cl)

**Abstract.** In semi-arid Chile, rock glaciers cover a surface area that is four-times larger than that occupied by glaciers. For this reason, their role in freshwater production, transfer and storage is likely to be of primary importance, especially in this area of increasing human pressure and high rainfall variability. To understand their hydrological role now and in the future it is necessary to characterize their internal structure (e.g., internal boundaries, ice, air, water and rock content). In this paper,

we present the results and interpretations of electrical resistivity and refraction seismic tomography profiles on an active (El Ternero) and inactive (El Jote) rock glacier in the Chilean Andes. These are the first in situ measurements in Estero Derecho: a natural reserve at the headwaters of the Elqui River, where the two rock glaciers are located. Within our study, we highlight the strong differences in the geophysical responses between active and inactive rock glaciers through the analysis and comparison of three different inversion schemes: individual dataset inversion, structural and petrophysical joint inversion. Moreover, we

propose a diagnostic model representation for the differentiation between active and inactive rock glaciers.

## 1 Introduction

High mountain environments are delicate geosystems under increasing human pressure and represent important geoecological indicators of a changing climate of the areas where they are situated. In particular, in semi-arid Chile (between 29°and 34°S), rock glaciers cover a surface area that is four-times larger than that occupied by glaciers (Azòcar and Brenning, 2010) and

likely play an important role in the hydrological cycle (Halla et al., 2020). While melt from seasonal snow cover provides a greater contribution to annual streamflow than rock glaciers, the snow cover only lasts 1 to 2 months after the last winter snowfall (Favier et al., 2009), so glaciers and other ice bodies are the main water source when river levels are at a minimum, especially during dry years (Gascoin et al., 2011). A recent study quantifying the contribution of rock glaciers to streamflow in the La Laguna Catchment (Elqui Province) indicated that the contribution at the end of summer may be significant (Schaffer

et al., 2019).

Rock glaciers are normally lobate or tongue-shaped landforms composed of rock fragments, sediment, ice, often unfrozen pore water, and contain air filled pore spaces and cavities (Cogley et al., 2011; Hauck et al., 2011). They are the visible



expression of the deformation of ice-rich creeping mountain permafrost and can act as climate change indicators in high
mountain environments (Barsch, 1989; Bodin et al., 2010; Berthling, 2011). Rock glaciers can be classified according to
the speed at which they move down slope through the deformation of subsurface ice and or ice-rich sediments (Ballantyne,
2002). Active rock glaciers contain enough buried ice to induce internal deformation and creep at a relatively fast pace, whereas
inactive rock glaciers contain less ice and move very slowly or are stagnant (Barsch, 1996; Brenning et al., 2007; Schaffer et al.,
2019). Because of their debris cover, rock glaciers are generally more resilient to climate (atmospheric) changes (Jones et al.,
2018; Harrington et al., 2018), although there are indirect measurements (e.g. a significant increase in solute concentrations for
rock glacier-fed lakes, increased velocities) which suggest that rock glaciers in the European Alps have experienced increased
melt rates in recent decades (Krainer and Mostler, 2006; Thies et al., 2007).

Studies on Andean rock glaciers (Schrott, 1996; Croce and Milana, 2002; Schaffer et al., 2019) indicate that they store
significant amounts of water and emphasise their role in freshwater production, transfer and storage. A study of the Tapado
Glacier complex, composed of a debris-free glacier, debris-covered glacier and a rock glacier in the Elqui watershed of the
Chilean Andes (Pourrier et al., 2014), describes the contrasting hydrological output of each formation. While water from
the debris-free glacier was highly correlated with daily and monthly fluctuations in temperature and solar radiation, the rock
glacier downstream buffered this variability acting as a reservoir during high melt periods and supplying water downstream
during low melt periods. This slow diffuse transfer of water was attributed to a highly capacitive but weakly transmissive
medium composed of a heterogeneous mixture of ice and rock debris. Harrington et al. (2018) investigated the contribution of
an inactive rock glacier in Canadian Rockies to the basin stream-flow. At this site, the rock glacier surface layer is composed
of coarse blocky sediments that allow the rapid infiltration of snow-melt and rain water which accumulates within the rock
glacier. Discharge from the rock glacier is slower than from the surrounding landscape and this rock glacier is therefore able
to provide up to 100 % of the basin streamflow during summer base-flow periods despite the fact that it drains less than 20
% of the watershed area. These two studies highlight the very important role rock glaciers play in moderating discharge to
ensure sufficient water reserves exist when river levels are at a minimum. This role is a critical one in semiarid Chile, an area
exposed to highly variable rainfall and little to no precipitation during the warmest months of the year (Garreaud, 2009; Valois
et al., 2020a), resulting in water scarcity especially at the end of summer compounded by growing economic demands for water
(Oyarzùn and Oyarzùn, 2011).

Understanding the distribution of ice, sediment, rock, water and air within a rock glacier is a critical first step for being
able to understand the way in which water flows through the landform and model its hydrology. Knowing the amount of water
stored (in liquid or frozen form) within a rock glacier is needed to estimate the water reserve available, which is particularly
important in the context of a warming climate. To estimate the volume a rock glacier occupies, the lower (bottom of the rock
glacier or bedrock geometry) and upper limits (bottom of the active layer or depth to permafrost) must be known. In addition,
since only part of the rock glacier is composed of ice, the percentage of ice must be quantified. This can vary considerably,
normally ranging from 40 % to 70 % in active rock glaciers (Barsch, 1996; Monnier and Kinnard, 2015).

In order to derive all the information mentioned above, we can rely either on direct observations (e.g., boreholes logs,
outcrops, tunnels and temperature measurements) or on surface-based geophysical observations (Hausmann et al., 2007).





Surface-based geophysical methods represent a cost-effective approach to investigate the physical structure and properties of the Earth's subsurface. Their ability to provide information over large areas with relative high resolution and in a non inva-
sive manner makes them a useful tool for studying ground ice and permafrost in high mountain environments, where difficult site access limits the possibility of deep borehole drilling (Maurer and Hauck, 2007; Hauck and Kneisel, 2008). For these reasons, geophysical methods have been used extensively to investigate the internal structure of rock glaciers (Hauck and Kneisel, 2008) and other geoforms such as high altitude wetlands (Valois et al., 2020b). Among the different techniques, the most implemented include refraction seismic tomography (RST), electrical resistivity tomography (ERT), ground-penetrating radar
(GPR) and gravimetry (Langston et al., 2011; Maurer and Hauck, 2007; Colucci et al., 2019; Pourrier et al., 2014).

A general issue with geophysical observations is that their information content is limited: even if geophysical surveys might result in vast datasets, only a finite number of model parameters can be independently inferred from them (Backus and Gilbert, 1970). To reduce this inherent ambiguity, complementary datasets taken at the same site can be incorporated and interpreted together. Joint inversion has become a popular tool in geophysics, providing a formal approach to integrate multiple datasets
with the aim of better constraining the model results (Moorkamp et al., 2016). In order to do that, the property models related to the different data sets need to be coupled, either through petrophysical relationships (Wagner et al., 2019; Mollaret et al., 2020) or by structural constraints (Jordi et al., 2019).

The goal of this study is to identify the geophysical signature differences between an active and inactive rock glacier, as well as testing several inversion methodologies to characterize their internal structure. In this paper, we present the results of
the characterization of an active (El Ternero) and inactive (El Jote) rock glacier located in the Chilean Andes. On both rock glaciers we conducted coincident RST and ERT profiles that we interpret both separately and jointly through two different schemes: the structural joint algorithm developed by Jordi et al. (2019) and the petrophysical four phase inversion scheme by Wagner et al. (2019). The geophysical profiles we took are the first in situ measurements over rock glacier in the reserve (Estero Derecho) where the two formations are located. Through the analysis and comparison of the inversion model results,
we were able to underline the response differences of the geophysical methods between active and inactive rock-glaciers and to infer key information regarding the subsurface structure and composition of the two glaciers. Moreover, from the analysis and comparison of the different inversion results it was possible to highlight the strong differences between the geophysical responses of the active and inactive rock glacier and to built a diagnostic model representation for the differentiation between active and inactive rock glaciers.

## 85 2 Study Area

The study area is in north-central Chile ($\sim 30°$ S). Here there is a sharp altitudinal gradient between the Pacific Ocean and the Andes mountains with peaks rising above 6000 m a.s.l. less than 150 km east of the ocean. At this latitude there exists intensive compression between the Nazca and South American tectonic plates, associated with a flat slab segment, which has resulted in the creation of major transverse valleys (Yáñez et al., 2001) such as the Elqui Valley in the Coquimbo Region (Fig. 1a).
The floor and marginal terraces of the Elqui Valley are of Quaternary alluvium. Surrounding mountains are steep and mostly





intrusive with some volcanic, volcano-sedimentary, and metamorphic rocks Paleozoic-Triassic in age (Aguilar et al., 2013; Valois et al., 2020a). Above 3000 m a.s.l. the mountain landscape has been carved by glaciers and typical geomorphology including U-shaped valleys, frontal and lateral moraines can be observed. Rock glaciers and periglacial landforms are numerous, particularly above 4000 m a.s.l.

The study site is within the semiarid Andes of Chile at the southern edge of the Arid Diagonal and Atacama Desert (Sinclair and MacDonell, 2016). Specifically it is located at the headwaters of the Elqui River within the Coquimbo Region in a nature reserve called Estero Derecho (Fig. 1b). In the city of La Serena on the coast the annual precipitation is $\sim 90 \text{ mm a}^{-1}$ (average from 1981-2016; Valois et al., 2020b), drastically lower than central and southern Chile. At the same time, demand from the agricultural sector, mining industry, and municipal water supply are high and water allocation has already been exhausted
here (de Agua., 2016). Precipitation increases with elevation reaching $\sim 160 \text{ mm a}^{-1}$ at 2900m a.s.l. in the Estero Derecho valley (Valois et al., 2020b). Increased precipitation at higher altitudes allows for the formation of a seasonal snow pack that completely melts during the spring and summer seasons (Réveillet et al., 2020). Variability in precipitation at an inter-annual time scale is linked to El Niño Southern Oscillation (ENSO; Favier et al., 2009), while at a decadal time scale it is linked to the Pacific Decadal Oscillation (Núñez et al., 2013). Precipitation has decreased since 1870 by $-0.52 \text{ mm a}^{-1}$ at La Serena. The
mean annual air temperature at a station at 3020m within Estero Derecho was 6.7°C between 2016-2020.

    Within the nature reserve there are no debris-free glaciers, only rock glaciers and periglacial landforms. The two rock glaciers measured in this study are locally known as "El Jote" and "El Ternero" and are in the eastern part of the nature reserve (Fig. 1b) at 3700-3870 m and 4170-4510 m a.s.l., respectively. El Ternero is the largest active rock glacier within Estero Derecho, has a lobate shape and obvious flow features including ridges and furrows, a steep frontal talus slope ($\sim 40°$), and well defined
lateral margins. There are a number of depressions $\sim 5$ m deep on the surface and a supraglacial pond covering an area of $\sim 80 \text{ m}^2$. El Ternero is 1.93 km long, has a maximum width of 0.51 km, and an area of 0.60 km$^2$. It is moving at a rate of $\sim 1 \text{ ma}^{-1}$ and lowering by $\sim 0.2 \text{ ma}^{-1}$ (based on three repeat differential GPS measurements taken in the summer of 2018-2019 and 2019-2020 between 4206 - 4417 m elevation). In contrast, El Jote has poorly defined flow features and a moderately sloping frontal slope ($\sim 24°$). This landform is inactive according to repeat differential GPS measurements taken taken at five
locations in the summer of 2018-2019 and 2019-2020. The lack of obvious flow features and its location within a cirque basin point toward the same conclusion. El Jote is 0.86 km long, has a maximum width of 0.48 km, and covers 0.31 km$^2$. Its surface is characterized by lobes as well as signs of subsidence such as depressions.

    At El Ternero a stream passes adjacent to the former and eroded terminal moraine of the glacier. The waterway initiates on the mountain slope above and south of the glacier and continues down-slope, eventually feeding a high altitude wetland and
the main waterway within the reserve, Estero Derecho. There is no evidence of water at the surface directly below the current frontal slope of the rock glacier. However, a substantial amount of water can be heard running underneath the rock glacier within topographic depressions on the surface and it is probable that this water feeds the stream at some point below the rock glacier. At El Jote water emerges $\sim 200$ m east of the main landform in a topographic low at $\sim 3740$ m a.s.l. It is unclear if this water originates from the rock glacier. There is a small periglacial feature directly above that may be contributing, but no
other obvious source of water visible at the surface. The waterway continues for $\sim 600$ m where it disappears $\sim 100$ m below



the frontal slope of the rock glacier. Water emerges in another, larger depression, along the same flow path $\sim 550$ m below the front of the rock glacier and continues down-slope, contributing to an alpine wetland (i.e. bofedal) and Estero Derecho. There is vegetation adjacent to the water; in contrast there is little to no vegetation in the surrounding landscape.

## 3   Methods

### 3.1   Geophysics measurements

Surface-based geophysical methods provide information about subsurface physical properties and have been extensively used to investigate the internal structure of rock glaciers. In particular, electrical resistivity and seismic refraction tomography are common choices for the characterization of rock glacier internal structure, even though their employment in mountainous environments demands specialised techniques for sensor coupling, data acquisition and inversion routines.

### 3.1.1   Electrical Resistivity Tomography (ERT)

ERT collects information about subsurface electrical resistivity ($\rho$) by injecting direct electric currents (DC) into the ground and measuring electric voltages at different locations. Data are obtained using a large number of resistance measurements made from spatially-distributed four-point electrode configurations (Binley and Kemna, 2005). The geometry of the current injection and potential electrode pairs are varied with typical set-ups involving many tens of electrodes and several hundred or thousand datapoints. These data are then inverted to infer the spatial distribution of electrical resistivity in the subsurface (Dahlin, 1996).

   Electrical resistivity quantifies how strongly a material opposes the flow of electric current. In most rocks and soils, electrical current is carried by movements of ions in the pore water (electrolyte conduction), with the actual mineral matrix practically acting as an insulator (Lesmes and Friedman, 2005). Due to the high contrast in resistivity between saturated and unsaturated sediments, and the marked increase of resistivity values at the freezing point, resistivity techniques have been useful in both hydrology (de Lima, 1995; Daily et al., 1992; Valois et al., 2018b, a) and permafrost-characterization studies (Evin et al., 1997; Hauck et al., 2003; Langston et al., 2011). In periglacial environments, the use of ERT is particularly popular due to the contrasting electrical resistivity corresponding to lithological media, water (highly conductive) and ice (which is assumed to be an electrical insulator). In Table 1 we list the relevant values for electrical resistivity in rock glacier environments, compiled from the literature.

   The main limitation for ERT is the need for the electrodes to have a good galvanic contact with the ground. Its application within the surveys was therefore problematic due to the heterogeneous and rocky ground surface which resulted in extremely high contact electrical resistivity. Following Maurer and Hauck (2007) methodology, we attenuated this problem by both facilitating the injection of electric current into the ground by attaching sponges soaked in salt water to the electrodes, and in addition, increasing the measured voltage by implementing the Wenner-Schlumberger array configuration (its low geometrical factor provides larger measured voltages compared to other options).





### 3.1.2 Refraction Seismic Tomography (RST)

RST is based on the analysis of first arrival traveltimes of critically refracted seismic waves to reconstruct seismic P-wave (i.e., compressional wave) velocity models (White, 1989; Nolet, 1987). When seismic waves impinge on velocity boundaries, they change their direction of propagation. At a critical angle that depends on the velocity contrast, head waves are created that
move along the interface at the speed of the faster lower-lying layer velocity and emits refracted waves. These refracted waves are measured by the receiver and the timing of their arrival (i.e., first-arrival travel times) are the main observations used in seismic refraction surveys.

Seismic velocity is the rate at which seismic waves propagate through rocks and soils and generally increase with the density of the material. In periglacial environments the large range of observed velocity values (Table 1) is favourable to the application
of RST, since large velocity contrasts between the underlying materials are necessary in order to effectively interpret the data. For this reason seismic refraction has been successfully used on rock glaciers since the 1970s (Barsch, 1971; Potter, 1972). In the last two decades the method has been extensively utilized in permafrost studies (Hauck et al., 2004; Mühll et al., 2002; Draebing and Krautblatter, 2012) and to monitor hydrodynamic variation impacts on velocities (Valois et al., 2016).

One limitation of first-arrival refraction methods is that they only use a small portion of the information contained in the
seismic traces and strongly depend upon the assumption that velocity increases with depth. In the case of velocity inversion (i.e., the deeper medium presenting a lower P-wave velocity than the overlaying one), the refracted wave will bend towards the normal. This gives rise to the so-called "hidden layer" phenomenon (Banerjee and Gupta, 1975). Moreover, surface conditions on rock glaciers highly attenuate seismic energy and make it difficult to couple geophones and seismic sources to the ground. During the collection of seismic data, we were able to partially improve the coupling through the use of a few geophones
fastened to metal plates. We also increased the signal-to-noise-ratio by repeating and stacking the same source position multiple times.

### 3.2 Acquisition strategy

Field data collection was conducted during the austral Summer: between the end of January and the beginning of February, 2020. The field campaign was logistically challenging in terms of both transportation of equipment and personnel to the site
and difficulty moving the sensors along the profile lines because of the extremely rugged surface of the rock glaciers and the altitude.

The location of sensors and sources of all the profiles were taken with a Trimble differential GPS. At both sites, we acquired the ERT surveys using Syscal Junior switch-48 (IRIS instrument, France) with 48 electrodes spaced 5 m apart and a Wenner-Schlumberger configuration. For El Jote rock glacier, the profiles length was 690 m (Fig.1c and d), that we obtained using five
sequential roll-alongs in which 50 % of the electrodes stayed in place each time and the other 50 % were displaced along the profile line. In total we implemented 144 different electrode positions and obtained 2135 measurement points. For El Ternero the profile length was 575 m (Fig.1e and f), which was obtained with four sequential roll-alongs. Here we used 120 different electrode positions and obtained 1479 measurement points. We recorded the refraction seismic surveys on both rock glaciers





implementing a Geode Exploration Seismograph device (Geometric, USA) along the same lines as for the ERT profiles. The
seismic source was a sledge hammer of 15 kg striking on a steel plate and we repeated each shot position five times in order
to improve the signal-to-noise ratio. For the profile taken on El Jote, we used 48 geophones with a spacing of 5 m and shots
in between geophone positions, but spaced 10 m apart. To obtain the length of 690 m we applied five sequential roll-alongs
as done for the resistivity line. In the case of El Ternero, the same spacing and configuration was used for both shots and
geophones, but after the first line, the failure of one of the cables reduced the number of geophones we could use to 24. The
total length of 575 m was then obtained by moving the 24-channels set-up four times and adding off-line shots to link the
different acquisitions. For both profiles, we manually picked the first arrival travel times on each trace resulting in 4575 picks
for El Jote and 1400 for El Ternero. For both datasets the error models resulted in 1 % relative error on the ERT observations,
which was obtained from the maximum of model reciprocity, and an absolute error of 0.001 seconds for the travel times,
estimated from the average variability of first arrival picking. The acquisition settings are summarised in Table 2.

## 3.3 Joint inversion

Geophysical methods only provide indirect information regarding the Earth's subsurface properties, so in order to interpret
them, the observations need to be inverted. With the aim of reducing the inherent ambiguity of the observations, aside from
the individual inversion, we decide to jointly interpret the RST and ERT data sets by implementing two alternative approaches:
structural and petrophysical joint inversion. All the inversion algorithms we use are written within pyGIMLI, an open-source
library developed in python for geophysical inversion and modelling which incorporates tetrahedral/triangular unstructured
meshes for model discretization (Rücker et al., 2017). In each case, to quantify the models ability to explain the field observa-
tions we refer to the root-mean square error (RMSE) and the error-weighted chi-square fit, where RMSE=0 and $\chi^2 = 1$ signify
a perfect fit (Günther et al., 2006).

### 3.3.1 Structural joint inversion

Structural coupling is a flexible constraint that enforces structural similarity between the property models. The assumption
in this case is that the proposed models must have a similar structure since they are describing the same subsurface area, in
addition to explaining the different collected observations. This similarity can be imposed either through common interfaces
(de Pasquale et al., 2019) or similar model gradients (Gallardo and Meju, 2004). Jordi et al. (2019) developed a structural joint
inversion algorithm that is suitable for 2D and 3D joint inversions on irregular meshes. The innovation in their approach is
the use of a cross-gradient operator based on a correlation model acting on physical length scales, and therefore reducing the
dependency on a particular mesh discretization. Their cross-gradient computation is an adaptation of Levièvre and Farquharson
(2013) for calculating the local spatial gradients in the model updates, where they calculate the model gradient at the location
of a cell by fitting a linear trend through the model parameters of that cell and its neighbourhood. The innovation of Jordi et al.
(2019) work stands in the definition of such neighbourhood, which is given as all cells in the model that lie within a predefined





distance. Their inversion scheme minimizes the following objective function:

$$\Phi = \Phi_{\mathrm{d}} + \lambda\Phi_{\mathrm{m}} + \lambda_{\mathrm{cg}}\Phi_{\mathrm{cg}} \tag{1}$$

where $\Phi_{\mathrm{d}}$ is the combined data misfit term, $\lambda$ and $\lambda_{\mathrm{cg}}$ are the geostatistical regularization and cross gradient weights, respectively. $\Phi_{\mathrm{m}}$ is the geostatistical regularization (Jordi et al., 2018), which can be thought of as superposition of damping and smoothing constraints (Maurer et al., 1998). In the case of 2D problems, the main parameters to be set for this operator are the

correlation lengths in the x and z directions ($\mathrm{I_x}$ and $\mathrm{I_z}$), defining the distance over which cells are considered to be correlated and the angle of rotation for the correlation model ($\Theta$) with respect to the Cartesian plane, so that the rotated coordinate axes can be set parallel to the geologically dominant directions. The last term, $\Phi_{\mathrm{cg}}$, is the cross-gradient operator and it is defined on the same scale as the regularization term.

In order to apply the structural joint inversion scheme, we first verify that the results from individually inverted data-sets

were not completely different, so that there is a structural similarity (Linde et al., 2008).

### 3.3.2 Petrophysical joint inversion

Petrophysical coupling allows the inversion of separate data sets to determine common parameters through petrophysical relationships. Within this framework, Wagner et al. (2019) developed an inversion routine which allows the interpretation of seismic refraction traveltimes and apparent resistivities in terms of ice, water, air and rock content. The inversion is based on

a petrophysics four phase model (4PM; Hauck et al., 2011), where permafrost systems are assumed to be comprised of the volumetric fractions of the solid rock matrix ($\mathrm{f_r}$) and a pore-filling mixture of liquid water ($\mathrm{f_w}$), ice ($\mathrm{f_i}$) and air ($\mathrm{f_a}$):

$$\mathrm{f_r} + \mathrm{f_w} + \mathrm{f_i} + \mathrm{f_a} = 1. \tag{2}$$

The treatment of the rock volumetric fraction as a single phase is a justified simplification in rock glacier environment, where the amount of soil is negligible compared to the hard rock.

The volumetric fractions in Eq. (2) are related to the seismic slowness (s), reciprocal of the P-wave propagation velocity (v), through the time averaging equation (Timur, 1968):

$$\mathrm{s} = \frac{1}{\mathrm{v}} = \frac{\mathrm{f_r}}{\mathrm{v_r}} + \frac{\mathrm{f_w}}{\mathrm{v_w}} + \frac{\mathrm{f_i}}{\mathrm{v_i}} + \frac{\mathrm{f_a}}{\mathrm{v_a}}, \tag{3}$$

and to the electrical resistivity through a modification of Archie's second law (Archie, 1942):

$$\rho = \rho_{\mathrm{w}}(1 - \mathrm{f_r})^{-\mathrm{m}} \left( \frac{\mathrm{f_w}}{1 - \mathrm{f_r}} \right)^{-\mathrm{n}}, \tag{4}$$

where the porosity is expressed in terms of rock content ($\phi = 1 - \mathrm{f_r}$). The assumptions within this 4PM model are that the medium is isotropic, at high effective pressure and has a single homogeneous mineralogy (validity of Eq. 3), and that the electric current flow is dominated by electrolyte conduction (validity of Eq. 4).

The petrophysical joint inversion scheme minimizes the following objective function (Wagner et al., 2019; Mollaret et al., 2020):

$$\Phi = \Phi_{\mathrm{d}} + \lambda\Phi_{\mathrm{m}} + \lambda_{\mathrm{p}}\Phi_{\mathrm{p}}. \tag{5}$$





Similarly to Eq. (1), the first term refers to the combined data misfit, while $\mathrm{Phi_m}$ represents a smoothness regularization term build through four first-order roughness operators to promote smoothness in the distribution of each constituent of the four-phase system. The last term is an additional regularization term which constrain the volume conservation (Eq. 2). The two weights $\lambda$ and $\lambda_\mathrm{p}$ are responsible for scaling the influences of the two regularization terms, where $\lambda$ is chosen to fit the data within the error bound and $\lambda_\mathrm{p}$ is chosen large enough to prohibit no-physical solutions.

Within this framework, the RST and ERT observations are used to infer the volumetric fractions of water, ice, air and rock for each model cell, while the spatial distribution of electrical resistivity and P-waves velocities are obtained through Eq. (3) and Eq. (4), where the petrophysical parameters and constituent velocities are assumed to be spatially constant. We chose the values for the inversion of the field observations based on the literature (Hauck and Kneisel, 2008; Maurer and Hauck, 2007; Hauck et al., 2011; Wagner et al., 2019), which are listed in Table 3. A last important parameter to consider in this scheme is porosity initial value and range. Wagner et al. (2019), already stressed the importance of a good porosity estimation in order to avoid ambiguity between ice and rock content and in a recent study, Mollaret et al. (2020) analyses the influence of the porosity constraint in the petrophysical joint inversion results. Following the approach of this last paper, and accordingly to the previous knowledge from the field site we tested different initial porosity values and ranges ($\phi_{min}$-$\phi_{max}$) for both rock glaciers. The choice was made selecting the less constraining intervals which allowed results consistent with the geology of the two sites.

## 4  Results

### 4.1  El Jote

Within the profiles taken on El Jote, the electrode positions coincide with the geophone locations. To interpret the observations obtained though seismic and resistivity tomography, we implemented individual dataset inversion and both petrophysical and structural joint inversion schemes as described in the Methods section. In all cases, we set a homogeneous resistivity starting model, with a value equal to the median of the apparent resistivities ($\rho\mathrm{a_{median}} = 4561\ \Omega\ \mathrm{m}$) and a gradient model for the seismic velocity, starting with $300\ \mathrm{m\ s^{-1}}$ at the top of the domain and gradually increasing to $5000\ \mathrm{m\ s^{-1}}$ at the base. The regularization weight was set to $\lambda = 10$ for all inversion schemes, chosen according to L-curve analysis (Hansen, 2001). As introduced in the methodology session, we firstly invert individually the data-sets in order to ensure structural similarity. The model results for El Jote are given in Fig. 2(a) and (b).

For the structural joint inversion scheme, we set the vertical and horizontal correlation lengths for the geostatistical regularization to $I_\mathrm{x} = 40\ \mathrm{m}$ and $I_\mathrm{z} = 15\ \mathrm{m}$, respectively, and the angle of rotation to $\Theta = -18.8°$ (equal to the dominant profile inclination). In addition, we set the cross gradient weight to $\lambda_\mathrm{cg} = 1000$ after L-curve analysis of the joint $\chi^2$ values. We obtained an overall data fit of $\chi^2 = 1.49$ after 15 iterations with RMSE for the ERT and traveltime data of 1.62 % and 1.55 %, respectively. Figure 3 displays the resistivity and velocity model results for the structural joint inversion scheme. At the top of the domain the model results show low velocity ($v < 10^3\ \mathrm{m\ s^{-1}}$) and high resistivity ($\rho > 10^4\ \Omega\ \mathrm{m}$), notably at approximately 300 m along the profile line, where the bulk of high resistivity values are concentrated and velocities are at a minimum. We interpret this layer as unconsolidated to highly fractured rocks with air filling pore spaces. This is coherent with field observa-



tions, where boulders are visible at the surface and possibly extend downwards along with unconsolidated rock till to depths
of 10 to 50 m. At the bottom of the domain the velocity model presents high velocity values between 150 m and 250 m and
at approximately 550 m along the profile line. In the first case the resistivity values are relatively low ($\rho \sim 10^3$ $\Omega$ m) while at
around 500 m they increase one order of magnitude ($\rho \sim 10^4$ $\Omega$ m). This increase can be explained by a decrease in the pore
space, where subsurface material between 150 m and 250 m is in liquid form (low resistivity) while near 550 m it is partly
frozen (high resistivity). These results are similar to the individual inversion results, but present an overall decrease in the
model parameters ranges in case of structural inversion.

For the petrophysical joint inversion scheme we applied the same regularization weight, as for the structural joint inversion:
$\lambda = 10$, while for ensuring the volume conservation we applied $\lambda_\mathrm{p} = 10000$. Regularization weights were chosen as illustrated
by (Mollaret et al., 2020), considering both, classic L-curve analysis and the sum of the components fractions. The initial
porosity was set to an homogeneous 30 % and inverted for in a range going from 0 % to 80 %, heterogeneously within
the model. After 15 iterations we obtained an overall data fit corresponding to $\chi^2 = 1.45$ with an RMSE for the ERT and
traveltime data of 1.42 % and 1.64 %, respectively. The results from the petrophysical joint inversion scheme are depicted in
Fig.4 and confirm the interpretation we gave for the structural joint inversion model outcomes and complement these results
with quantification of the different subsurface components. The top layer presents a large air content (up to 63 %, see Fig.
4e) and low rock fraction (with a minimum of 27 % at the surface, see Fig. 4f). Underneath, the unconsolidated rocks are
characterized by a decrease in porosity and relatively high content in water (up to 29 %, see Fig. 4c) except near the profile
length of 550 m, where the ice content slightly increases to 3 % (Fig. 4d). In addition, the high rock content at the bottom of
the domain (88 %, see Fig. 4f) likely represents the top of the bedrock. Besides the similarity in the structure and component
interpretation of the subsurface, the transformed velocity and resistivity models (Fig. 4a and b) present differences if compared
to the structural joint inversion results, with overall lower velocity values and higher contrasts in the resistivities.
In Fig. 5 we compared the relative residuals of the three inversion schemes computed as:

$$\mathrm{Res} = \frac{\mathbf{d}_\mathrm{obs} - \mathbf{d}_\mathrm{mod}}{\mathbf{d}_\mathrm{obs}}(\%), \tag{6}$$

with $\mathbf{d}_{OBS}$ standing for the field observations (ERT or RST) and $\mathbf{d}_{MOD}$ for the response of the inversion model results.

### 4.2 El Ternero

Collection of the profiles on El Ternero were logistically more complicated than on El Jote, due to higher altitudes and espe-
cially to the extremely heterogeneous, rocky surface. Even though resistivity and seismic lines were identical, some variation
can exist between geophone position (on a rock in case of flat geophone) and electrode position (between rocks). The position
differences are much lower than the spacing between two sensors, so that such positioning error should not have a significant
impact on the results. In order to implement the joint inversion schemes we therefore needed to adapt the sensor locations to
lie within the same line. Because RST is more sensitive to the local geometry than ERT, we decided to adapt the electrode
locations to the geophones line while preserving the relative distances of the electrodes. Overall, the data quality for this rock
glacier is much lower than for El Jote, which is impacted by the significant decrease in the number of data points for both





ERT (almost 1.5 times less than for El Jote) and RST (more than 3 times less than for El Jote, see Table 2). As for El Jote, we initiated the inversion schemes with a homogeneous resistivity, with values equal to the median of the apparent resistivities ($\rho a_{\mathrm{median}} = 36054\ \Omega$ m) and the same gradient model for the seismic velocity (i.e., 300 m s$^{-1}$ at the top and 5000 m s$^{-1}$ at

the bottom of the domain). We applied the same regularization and cross-gradient weights as for the other site, only changing the angle of rotation for the regularization operator to the dominant profile inclination for El Ternero: $\Theta = -10°$. As for the previous case, we firstly individually invert the observation to asses structural similarity (see Fig. 2c and d).

From the structural joint inversion, we obtain an overall data fit of $\chi^2 = 1.51$ after 15 iterations with RMSE for the ERT and traveltime data of 1.90 % and 0.91 %, respectively. In Fig. 6 we show the resistivity and velocity model inversion results

for the structural joint inversion scheme. The results show a thin layer (approximately 5 m thick) of low velocity and high resistivity which, as for El Jote, reflects the surface of the rock glacier: unconsolidated rock with air-filled pore space. Below this layer, P-waves velocity increases gradually for the first 15-20 m up to v $\sim 3000$ m s$^{-1}$ and has a sharp increase from 25 m depth (v $> 4000$ m s$^{-1}$). We interpret the gradual increase of velocity as a lowering in the pore space, and sharp changes with either the presence of intact rock (i.e., top of the bedrock) or with a significant increase in the amount of ice (i.e., top of the

permafrost layer). Also, at approximately 150 m and 450 m of the profile length the resistivity has two low-value anomalies which most likely reflect the presence of water within the pore space. Also, a water pool was observed at a distance of about 50 m perpendicular to the 150 m abscissa. Comparing these results with the individual inversion, we see that the general structure of the models is similar with an overall decrease in the model parameters ranges in case of structural inversion.

For the petrophysical inversion scheme, the initial porosity was set to an homogeneous 60 % and inverted within a range

going from 10 % to 90 %, heterogeneously within the model. After 15 iterations we obtained an overall data fit corresponding to $\chi^2 = 1.65$ with RMSE for the ERT and traveltime data of 1.78 % and 1.30 %, respectively. In Fig. 7 the joint inversion results obtained through petrophysical coupling are shown, which help to clarify the information gained through the structural joint inversion. Indeed, they confirm the presence of a thin top layer with moderately high fraction of air (up to 28 %, see Fig.7e) overlaying a layer with a high ice content (more than 30 % for the majority of the domain and up to 44 % at its highest

concentration (Fig. 7d) except near profile length 150 m and 450 m, where the fraction of water slightly increases to 15 % and 17 %, respectively (Fig. 7c). As in the previous case, the transformed velocity and resistivity models (Fig. 7a and b) present differences if compared to the structural joint inversion, with overall lower velocity values and higher resistivities values and contrasts. In Fig. 8 we compared the relative residuals of the three inversion schemes for El Ternero glacier (Eq. 6).

## 5   Discussion

### 5.1   Comparison of the inversion routines

The overall structure of the inversion model results is largely coherent, with the main features of high/low resistivities and high/low velocities presented in the individual inversion results preserved for both the joint inversion schemes. Nevertheless, from the comparison of the numerical values of velocity and resistivity, we can stress some common patterns related to the inversion routines. In case of single inversion results, P-wave velocity models (Figs. 2a and c) presents some unrealistic values





as extremely low velocity at the surface for El Jote ( $v_{min}$ 10 m s$^{-1}$ ) and extremely high velocity at the bottom of El Ternero ($v_{max}$ $10^4$ m s$^{-1}$). In the first case, the low values are compensated by a high velocity anomaly at the bottom of the domain which occupies a larger volume if compared with the results of both the joint inversion routines (Figs. 3a and 4a), while for El Ternero, the high values are counterbalanced by lower velocity at the surface ($v_{min}$ 100 m s$^{-1}$) if compared with the one obtained through joint inversion routines ($v_{min}$ 400 m s$^{-1}$ in case of structural joint inversion and $v_{min}$ 900 m s$^{-1}$ in case

of petrophysical joint inversion). Also, for both cases the petrophysical joint inversion results present the smaller ranges of P-wave velocities and the smoothest contrasts within the model. Regarding the resistivity model results, the smaller ranges and smoothest contrast within the model rises from the structural joint inversion (Figs. 3b and 6b), while the petrophysical coupling gives the highest values and sharpest contrasts within the model (Figs. 4b and 7b). Individual inversion in this case present the smaller minimum for the resistivity values for both rock glaciers and a larger range of velocities if compared to the structural

joint inversion results.

The structural joint inversion results are the closest in aspect and (absolute) numerical values to the individual inversion, of which resolves some ambiguity especially within the velocity models (the low values at the top of El Jote and the exceptionally high values at the bottom of El Ternero investigated domains disappears for the joint inversion results). Nevertheless, the interpretation of structural or individual inversion results in terms of subsurface components (i.e., air, rock, water or ice contents) is

difficult and prone to different interpretations. On the other side, the petrophysical joint inversion allows a direct estimation of these components. Still, the resistivity and velocity models presents a strong numerical difference if compared to the individual inversion results. Part of this discrepancy can be related to the choice of petrophysical relationships and parameters from which the physical properties are computed, which in these case where chosen based on literature because of the lack of previous knowledge. Also, a large part of such discrepancies are most probably part of the remaining ambiguity within the interpretation

of observations.

For both rock glaciers the structural coupling is able to slightly increase the data fit for RST observations, while the petrophysical coupling presents an improved ERT data fit (Table 4). In both cases, the RMSE obtained in the case of joint inversion increases only slightly if compared to the inversion of individual datasets, proving that it is possible to decrease the ambiguity of the inversion results without significantly losing accuracy for the model responses in describing the observations.

## 5.2 El Jote (inactive rock glacier)

For El Jote, the results show a top layer (laterally variable between 10 to 50 m thick) of unconsolidated rock with air-filled pore space, especially from 300 m to the end of the profile line. This overlays a layer where the pore space decreases and appears saturated with water for the majority of the line, apart from near 550 m, where the fraction of ice slightly increases to 3 % (Fig.4d). The increased velocity and resistivity at 550 m could also be interpreted as the presence of intact rock, as opposed

to ice. Indeed, when the porosity of the subsurface is unknown, the petrophysical joint inversion scheme does not easily differentiate between ice and rock content (Wagner et al., 2019). In order to gain information about porosity, we unsuccessfully attempted to drill a core sample, but the drill broke at very shallow depths. Nevertheless, for both interpretation outcomes, the overall results are a reflection of the permafrost degradation on El Jote. It is likely that within this inactive rock glacier, the ice





has melted, leaving behind large voids filled with air (top layer) or water (deeper layer). For both the inversion results it seems
that the bedrock is deeper than 100 m for almost the entire profile length, but the strong increase in velocity (Fig. 3a) and rock
content (Fig. 4f) at the bottom of the domain and at profile lengths of 150 m to 250 m and at approximately 550 m may be
interpreted as the top of the bedrock. In addition, the high water content in the bottom layer (more than 20 %) suggests the
presence of an aquifer between the bedrock and the surface of the inactive rock glacier.

### 5.3  El Ternero (active rock glacier)

The inversion model results for El Ternero are slightly shallower than the ones obtained on El Jote. This is due to the extremely
irregular surface of this rock glacier which increased the dispersion of seismic energy and due to the failure of one of the two
geophone cables: the off-line shots used to link the displaced arrays were recorded only by few of the closest geophones to the
shot position, thereby losing ray coverage with depth. Nevertheless, we were able to retrieve useful information from the field
measurements. Both inversion scheme outcomes show a 5 m-thick active layer made of unconsolidated rock with air-filled
pore space, overlaying a layer where the percent of frozen material increases drastically and largely homogeneously within
the investigated profile, suggesting the presence of thick permafrost (25 m-50 m minimum, and probably continuing below the
maximum investigation depth). Considering the ice content from the petrophysical coupling (Fig. 7d) we would interpret the
top of the permafrost layer at 5m, but from structural inversion results (Fig. 6a) we observe a steep increases in velocity values
located between 10 m and 25 m depth, which is would most likely indicate rock compaction. Nevertheless this layer is not
continuous because of the low resistivity anomalies near 150 m and 550 m along the profile line, which are either a sign of
local melting and therefore of a degradation of the permafrost or of reaching of the bedrock (bottom of ice-rich layer).

### 5.4  Towards a diagnostic model representation for the ice or water presence in rock glaciers

Both joint inversion schemes give satisfactory results which are overall consistent with each other for El Jote, but leave some
ambiguity in case of El Ternero. Nevertheless these interpretation methods are not always applicable as the petrophysical
joint inversion may be limited by the lack of proper petrophysical models (or parameters), and the structural coupling may
not be suitable due to the inherent properties of the different data sets ( e.g., resolution, signal-to-noise ratio and sensitivity
to lithological structures). When model coupling is not possible, the comparison of velocity and resistivity model inversion
results can still deliver plenty of information about the rock-glacier's internal structure. For example, in Fig. 9 we show scatter
plots of resistivity vs velocity for (a) El Jote and (b) El Ternero individual (blue asterisks) and structural joint inversion results
(orange asterisks). In the first case, we can see for both inversion approach a bulk of points characterized by velocity lower
than 1000 $\mathrm{m\,s^{-1}}$ and relatively high resistivities ($\rho \sim 10^4 \Omega$ m) which correlates with the large air content modelled at the
top of the investigated domain by petrophysical coupling (Fig. 4e). The overall trend is represented with a solid line (red for
individual and black for structural joint inversion results), which has a negative coefficient of slope which is larger for the
structural joint inversion (m=-0.28) than for the individual inversion (m=-0.18). This means a decrease in resistivity (reduction
in air or increase in water content) while velocity increases (reduction of the pore space or consolidation with ice presence
or healthier rocks). This trend is coherent with the hypothesis of a subglacial aquifer, already deducted by the water content





obtained by petrophysical joint inversion (Fig. 4c). For El Ternero, the scatter plot shows a rather different behavior: a large density of points presenting high resistivity ($\rho > 10^4 \Omega$ m) and velocities between 1000 and 3500 m s$^{-1}$, followed by a general trend of increasing resistivity and velocity. Also in this case, the slope coefficient in case of structural joint inversion (m=0.43)

is larger than the one from individual inversion (m=0.27). Such a trend is consistent with the ice rich layer depicted in Fig. 7(d), with higher resistivities and velocities. Moreover, for both scatter plots the results from individual inversion are more disperse compare to the one from structural joint interpretation. The rather different appearance of the two scatter plots can be used as an indicator of the distinct nature of the two rock glaciers: overall, the inactive rock glacier is characterized by lower resistivities and a large range of velocities representing either air (v $< 1000$ m s$^{-1}$) or water filled pore space, while the active

rock glacier is indicated by higher resistivity and velocity values, reflecting the ice rich layer. Moreover, in the case of the active rock glacier the slope coefficient is positive, while it is negative in the inactive case. This type of model representation is a valuable tool for interpreting the inversion model results and can be implemented as well in case it is not possible to implement neither structural nor petrophysical coupling.

## 5.5  Hydrogeological role

El Ternero and El Jote represent two end-members of rock glacier types. El Ternero is an active rock glacier containing significant amounts of ice according to our geophysical analysis (Fig. 7d), while El Jote is likely an inactive rock glacier whose ice has largely if not completely melted to form an aquifer (Fig. 4c and d). Each have a distinct and important hydrological role. El Ternero has the capacity to function as a long-term aquifer given that most of the water it contains is in the form of ice which is insulated from the environment by debris cover ($\sim 5$ m thick). The insulating effect of the debris cover has been

shown to slow the rate of melt (Jones et al., 2018; Bonnaventure and Lamoureux, 2013) making rock glaciers more resilient to climate change compared to debris-free glaciers. Field observations and results from the joint inversion modeling suggest that some of the ice contained within El Ternero is melting. There are various oval-shaped topographic depressions $\sim 5$ m deep on the surface of the rock glacier including a supraglacial lake that is $\sim 80 \ m^2$. We interpret these depressions as thermokarst degradation features. In some of these depressions a substantial amount of water can be heard flowing which may be derived

from melting of ice within the rock glacier. The joint inversion results show that there may be water accumulating at 450 m along the profile (Fig. 7c).

El Jote has an important hydrological role as an aquifer and in terms of its influence on water flow. Assuming its impact on hydrology is similar to other rock glaciers for which measurements exist in the semiarid Andes and elsewhere, it is likely to act as a reservoir during high melt and/or precipitation periods then release the water downstream at a rate slow enough to

contribute significantly to streamflow when it is needed most at the end of summer (Pourrier et al., 2014; Harrington et al., 2018). It is possible that the surface streams down-glacier are the surfacing of water from the rock glacier aquifer, although this hydrological link has not been measured in the field. This role in water storage and delayed runoff is critical for the Coquimbo Region where annual precipitation is very low ($<100 \ mm \ a^{-1}$ at the coast), precipitation events are variable, and there is little to no precipitation during the warmest months of the year (Garreaud, 2009; Valois et al., 2020a).



## 6 Conclusion & Outlook

In this study, we presented the first comparison of geophysical signatures of one active and one inactive rock glaciers using independent, structurally coupled and petrophysical coupled inversion routines. We found that each routine agrees with much higher velocities and resistivities for the active rock glacier, which are interpreted as a much higher content of ice in the petrophysical model. Scatter plots of velocity versus resistivity point out as well a clear signature difference for both rock glacier, which could be interpreted as the difference between a relict glacier with an aquifer and an active glacier containing ice-rich layers.

The analysis of the inversion results confirms the effectiveness of geophysics for rock glacier (active or inactive) characterization but at the same time proves the necessity for complementary geophysical measurements: only velocity or resistivity information does not allow the differentiation between the distinct subsurface components (i.e., air, water, rock and ice). The RMSE linked to the data fit of the joint inversions only increases slightly if compared to the inversion of individual datasets, proving that it is possible to decrease the ambiguity of the inversion results without significantly losing accuracy for the model responses in describing the observations.Both joint inversion schemes could be used on their own to interpret the observation, nevertheless the comparison between the results of both schemes helped us to reduce some of the model results ambiguity.

Through the joint interpretation of ERT and RST surveys for El Jote we were able to characterize its inner structure, detect the top of the bedrock in part of the model domain and identify a potential aquifer, while in case of El Ternero we could clearly determine the active layer and the top of the permafrost, together with the sign of its partial melting at the bottom of the sensed domain. Nevertheless,there is ambiguity in the interpretation between ice and rock matrix (especially for El Jote inversion results), which could be improved adding information about subsurface porosity or by the incorporation of additional freeze–thaw sensitive data sets such as complex electrical resistivity measurements (Wagner et al., 2019). In addition, to increase the investigated depth it would be necessary to improve the seismic data quality, which could be (time-expensively) done by fastening the geophones to the surface by drilling small holes in the rock.

*Acknowledgements.* The field campaign to obtain the geophysical profiles on the two glaciers was logistically and physically challenging because of the location and altitude. This data collection was possible thanks to: Eduardo Andréz Yáñez San Francisco, Gonzalo Alfredo Navarro Chamal, Marcelo Andrés Marambio Portilla, Ivan Fuentes, Jorge Sanhueza Soto, Benjamin Lehmann, Ayón García Piña, Christopher Leonardo Ulloa Correa and Ignacio Díaz Navarro. In particular we would like to thank Eduardo Yáñez and the U. Atacama Group for the tremendous help in the field. We also wish to thank Claudio Jordi for sharing his code and Florian Wagner and his research group for making it available to the public. We thank Ignacio Castro Cancino for his contribution to the rock glacier descriptions and for providing the MAAT data for the station in Estero Derecho. This work was supported by CONICYT-Programa Regional-Fortalecimiento (R16A10003) and FIC-R (2016) Coquimbo (BIP: 40000343). Nicole Schaffer was supported by CONICYT-FONDECYT-Postdoctorado (3180417).



480   *Author contributions.* GdP and RV analysed the data. GdP ran the inversions and wrote most of the manuscript except Sections 2 and 5.5, which were written by NS. RV, SMD and NS designed the study, organised the field campaign, reviewed and edited the manuscript. All authors contributed to the study

*Competing interests.* No competing interest are present



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



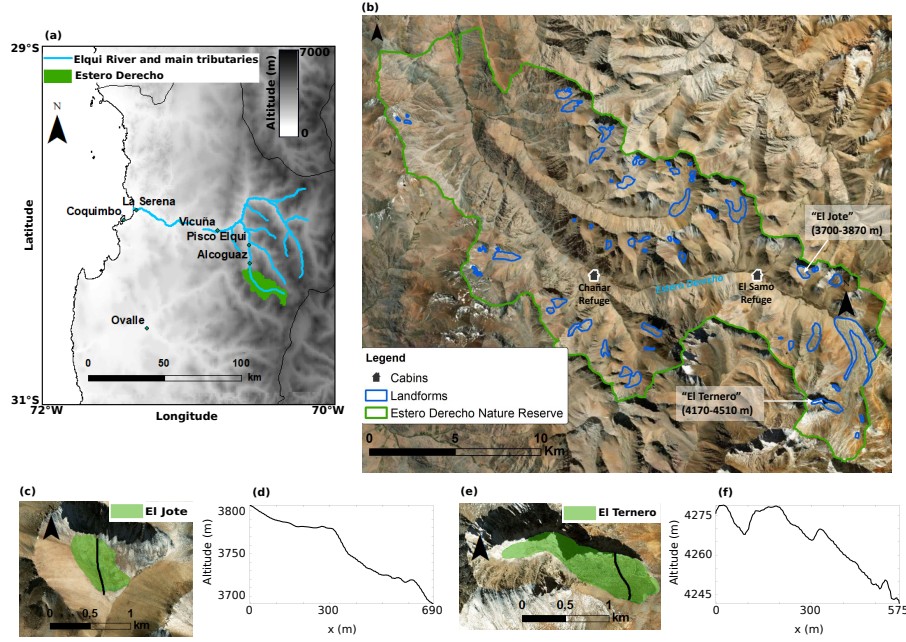

**Figure 1.** (a) Overview map indicating the location of Estero Derecho (∼ 30°S, 70°W ) in the Coquimbo Region of Chile. Elevation map from ASTER GDEM. (b) Detailed map of Estero Derecho with an inventory of landforms created by CEAZA. The delineations for El Jote and El Ternero were created specifically for this study from the Esri base-map satellite imagery. Both landforms are labeled with their respective elevation ranges. (c) Aerial image of El Jote, showing the location of the geophysical survey line and (d) its topography. (e) Aerial image of El Ternero, showing the location of the geophysical survey line and (f) its topography.



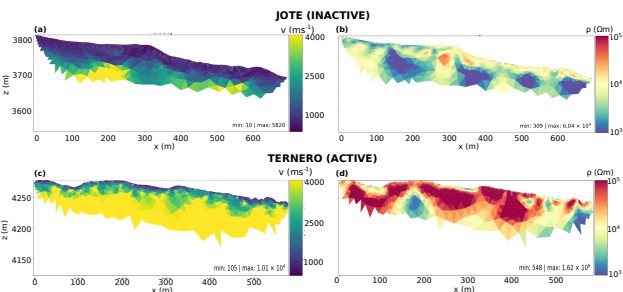

**Figure 2.** Single dataset inversion results. El Jote (a) velocity and (b) resistivity tomograms. El Ternero (c) velocity and (d) resistivity tomograms. The models are cut off below the lowermost ray path. The velocity colorbar is linear, while the resistivity one is expressed in logarithmic scale.



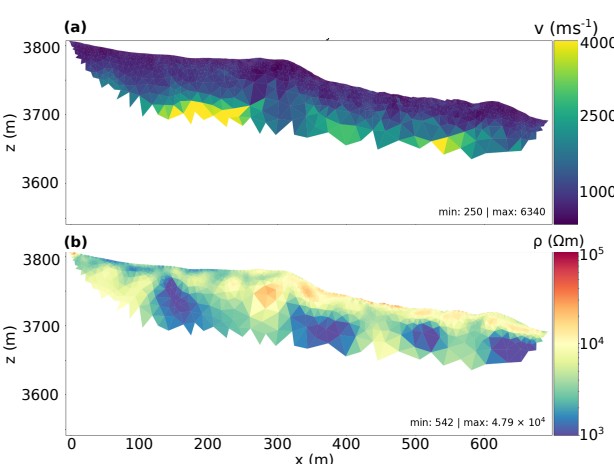

**Figure 3.** Structural joint inversion results of El Jote field data sets. (a) Velocity and (b) resistivity tomograms. Both models are cut off below the lowermost ray path. The velocity colorbar is linear, while the resistivity one is expressed in logarithmic scale.



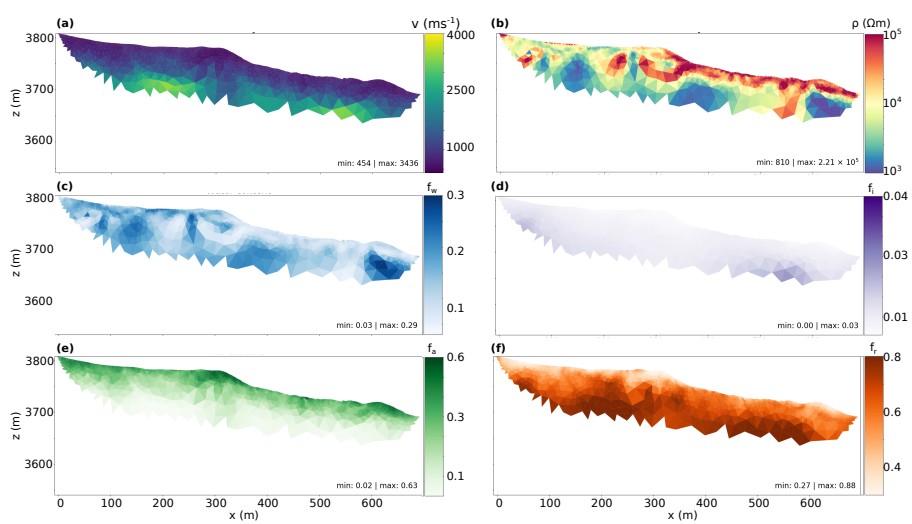

**Figure 4.** Petrophysical joint inversion results of El Jote field data sets. The tomograms represents (a) velocity and (b) resistivity transformed models. The directly inverted parameters are (c) water, (d) ice, (e) air and (f) rock content. All models are cut off below the lowermost ray path, with only resistivity colorbar expressed in logarithmic scale.





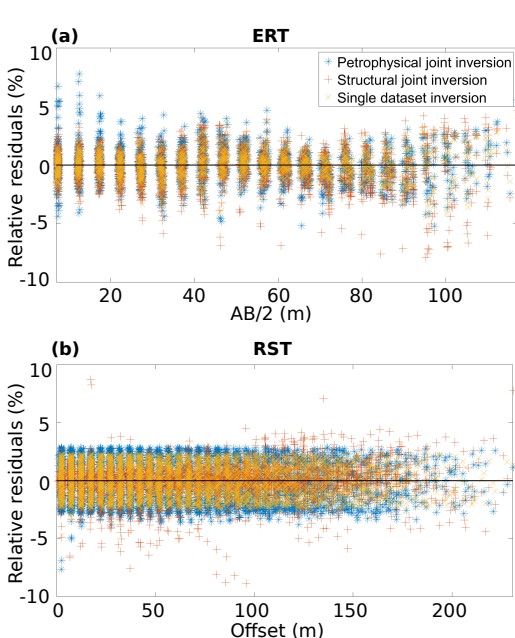

**Figure 5.** Relative residuals of (a) ERT and (b) RST individual and joint inversion results for El Jote glacier (Eq. 6).



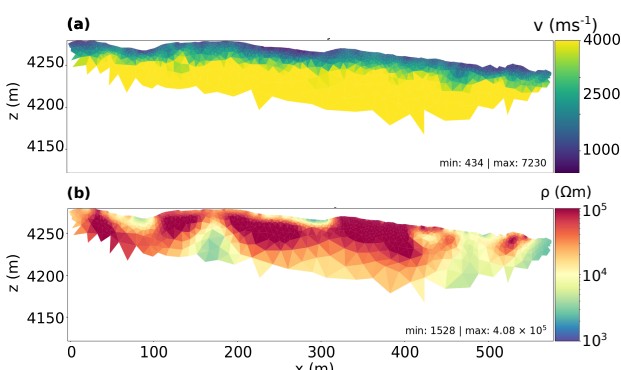

**Figure 6.** Structural joint inversion results of El Ternero field data sets. (a) Velocity and (b) resistivity tomograms. Both models are cut off below the lowermost ray path. The velocity colorbar is linear, while the resistivity one is expressed in logarithmic scale.



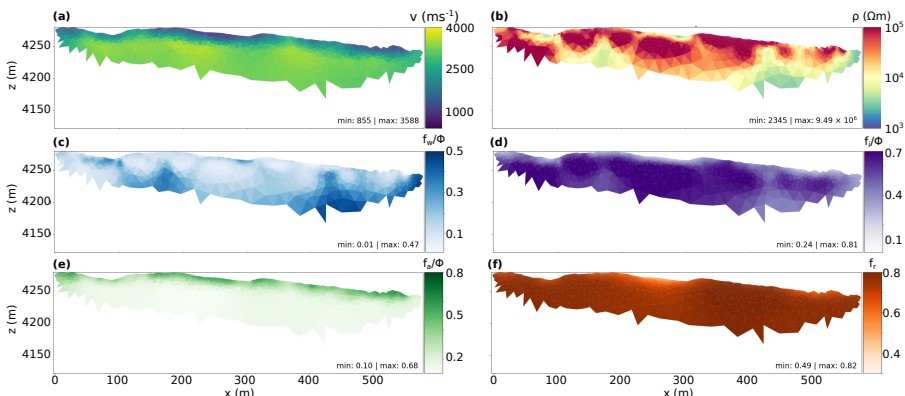

**Figure 7.** Petrophysical joint inversion results of El Ternero field data sets. The tomograms represents (a) velocity and (b) resistivity transformed models. The directly inverted parameters are (c) water, (d) ice, (e) air and (f) rock content. All models are cut off below the lowermost ray path, with only resistivity colorbar expressed in logarithmic scale.



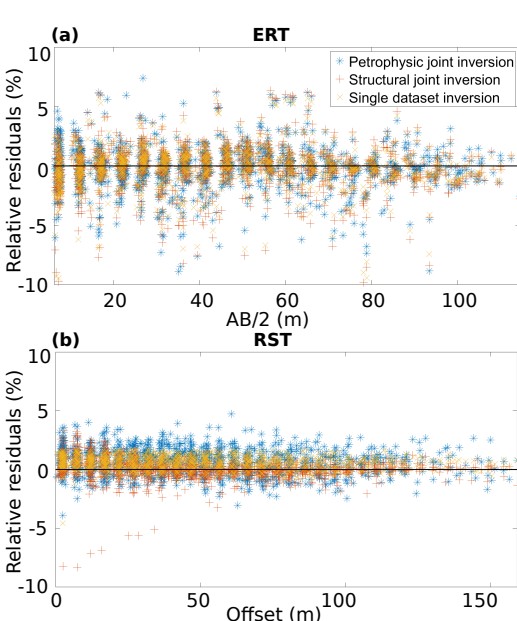

**Figure 8.** Relative residuals of (a) ERT and (b) RST individual and joint inversion results for El Ternero glacier (Eq. 6).





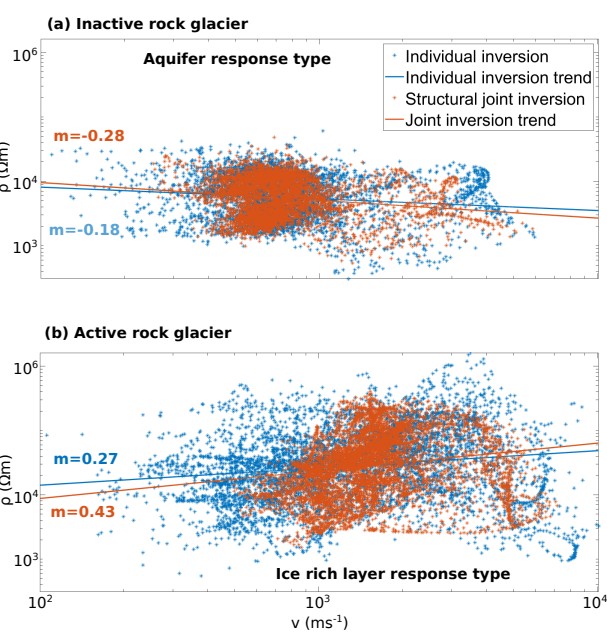

**Figure 9.** Scatter plot of resistivity versus P-waves velocity values for the individual and structural joint inversion of (a) El Jote and (b) El Ternero datasets. For both sites and inversion schemes we represents the linear trend of the plots (solid lines) together with their slope coefficient values (m).





**Table 1.** Relevant physical parameters for ERT and RST surveys on rock glaciers (table compiled from Hauck and Kneisel, 2008 and Maurer and Hauck, 2007).

|  | Electrical resistivity ($\Omega$ m) | P-wave velocity (m s$^{-1}$) |
|---|---|---|
| Sand-Gravel | $10^2 - 10^4$ | 400-2500 |
| Rock | $10^3 - 10^5$ | 3000-6500 |
| Glacial ice | $10^6 - 10^8$ | 3100-4500 |
| Frozen sediments, ground ice, permafrost | $10^3 - 10^6$ | 2500-4300 |
| Water | $10^1 - 10^2$ | 1500 |
| Air | $10^{14}$ | 330 |





**Table 2.** Acquisition settings for ERT and RST profiles on El Ternero and El Jote.

|  | El Jote | | El Ternero | |
|---|---|---|---|---|
|  | ERT | RST | ERT | RST |
| sensor positions | 144 | 144 | 120 | 120 |
| sensors spacing (m) | 5 | 5 | 5 | 5 |
| number of shots | - | 98 | - | 75 |
| shots spacing (m) | - | 10 | - | 10 |
| profile length (m) | 690 | 690 | 575 | 575 |
| data points | 2135 | 4575 | 1479 | 1400 |
| measurement errors | 1 % | 0.001 (s) | 1 % | 0.001 (s) |





**Table 3.** Parameters used for the petrophysical joint inversion of El Jote and El Ternero datasets (eqs.3 and 4).

| Archie parameters | | Constituent velocities | |
| --- | --- | --- | --- |
| $\rho_w$ | 60 ($\Omega$ m) | $v_w$ | 1500 (m s$^{-1}$) |
| n | 2.4 | $v_i$ | 3500 (m s$^{-1}$) |
| m | 1.4 | $v_a$ | 330 (m s$^{-1}$) |
| | | $v_r$ | 6000 (m s$^{-1}$) |





**Table 4.** RMSE between ERT ad RST observations and the modelled responses for El Jote and El Ternero inversion results with individual dataset inversion, structural and petrophysical coupling.

|  | El Jote | | | El Ternero | | |
|---|---|---|---|---|---|---|
|  | Individual | Structural | Petrophysical | Individual | Structural | Petrophysical |
| RST | 1.31 % | 1.55 % | 1.64 % | 0.83 % | 0.91 % | 1.30 % |
| ERT | 1.39 % | 1.62 % | 1.41 % | 1.68 % | 1.90 % | 1.78 % |