# Peer review of "Contrasting geophysical signature of a relict and an intact Andean rock glacier"

_The Cryosphere, 2020_

## Referee Comment (RC1) · Anonymous Referee #1 · 11 Dec 2020

Dear authors, dear Editor,

The manuscript "Active and inactive Andean rock glacier geophysical signatures by comparing 2D joint inversion routines of electrical resistivity and refraction seismic tomography" presents the application of seismic refraction (SRT) and electrical resistivity tomography (ERT) methods for the investigation of two block glaciers. The objective of the manuscript is to gain information about the internal structure of the glaciers, in particular the water and ice content. The authors use two different joint-inversion algorithms that permit to resolve for a subsurface model (define by physical properties,

i.e., electrical resistivity and changes in seismic velocities) that simultaneously explain the data collected with the two different geophysical methods. This is a study in line with the interest of "The Cryosphere". Although the use of SRT and ERT for the characterization of a rock glacier has been addressed before, the use of two joint-inversion algorithms offers an interesting perspective and make the study relevant beyond other case-studies. Thus, I recommend the publication of the manuscript. However, I think that the paper could be significantly improved in its structure and the actual discussion of the results. I am here attaching a marked .PDF where i pointed to some lines that I think need to be evaluated and corrected by the authors. There point to formulations that need to be improved to avoid misunderstandings or possible technical errors in their descriptions. However, while readings the manuscript I found eight major concerns, which I think the authors should address: 1) I feel the tittle to be too provocative and/or to certain extent misleading. I understand that the authors want to stress the comparison of results obtained through two different inversion algorithms. However, I am not sure that the comparison is valid taking into account that one refers to a spatial regularization schemes and the other one aims at solving a set of petrophysical models. Moreover, the joint-petrophysical inversion uses a set of calibration parameters (presented in Table 3 and Table 4) to permit the computation of air, ice and water content. However, in the present study, such parameters are taken from the literature, even if these are calibration values than need to be adjusted to each site (see the works from Archie and the reference to the studies by Glover et al., (2000) and Glover (2009) – full references below). In this regard, the comparison is unfair and technically limited. Would not be better to change the tittle to something like "extended interpretation based on the application of two joint-inversion algorithms"? 2) If the authors decide that the comparison of the results is relevant, then I suggest that the authors provide a quantitative comparison of the parameters obtained through the joint inversion, i.e., the seismic velocities and electrical resistivity resolved from the 2 inversion algorithms. Right now, the authors present only the plot of the inversion results and force the readers to compare those results by means of color-coded images in different pages and

sizes. I feel such comparison to be at best qualitative and open to debate. It would be better if the authors plot the parameters solved for both strategies (for example, Vp from the joint-petrophysical inversion vs. Vp from the structural joint inversion). In this case, deviations between both approaches could be quantified. Moreover, such analysis would be also convenient within a numerical study (with Gaussian error), where deviations from the truth model can be also quantified. 3) In order to perform a proper analysis of the two algorithms, the authors should investigate the variations in the retrieved models after testing different parameters used for the inversion. In this regard, the authors could investigate the resulting seismic velocities and electrical resistivity values after testing a few parameters in the petrophysical joint inversion and a few combination of the scale-length correlations for the joint-structural inversion. Right now, the study runs a set of inversions with some values extracted from the literature (for the petrophysical inversion), and based on the slope (for the structural inversion. Are we expecting the models to be comparable? – actually we are forcing the joint inversion to converge with some predefined settings that might not accurately describe the field conditions. In this regard, the users might be causing larger uncertainties in the inversion than just solving for a smooth-constraint independent inversion of the different data sets. I think that the proper comparison of the different joint-inversion algorithms needs to address the use of adequate parameters, or at least assess the deviations in the retrieved models by an inadequate selection of the inversion settings. The use of joint-inversion schemes has been largely investigated in geophysical studies, still they are not widely-accepted as they rely on the use of site-specific models or require of a correlation between the different parameters that might not exists. I think the authors point out to this problem. The use of a numerical study would be also a good option to extend the analysis to quantify deviations from the truth model. 4) Regarding the correlation of the seismic and electrical parameters, I find Figure 9 quite intriguing. Actually, the authors demonstrate no correlation between the seismic velocities and the electrical resistivity. I can see a cloud of points with a large variance and to pattern. However, the authors describe a correlation and quantify a model linking both of the

properties. However, the authors do not present the correlation coefficients that actually quantify the actual correlation. The actual lack of correlation observed in Figure 9 is especially disturbing taking into account that the use of the joint-structural constraints aims at improving such correlation. Is such poor correlation due to the inadequate correlation lengths selected in this study? Is this a problem of poor data quality? If the authors cannot address this question in detail, I think that the authors should completely remove Figure 9. If the authors decide to keep the figure, please write explicitly the correlation coefficient and address in detail the lack of correlation. 5) I would like to get further information regarding the reasons to select the correlation-lengths used in the joint-structural inversion. Did I understood correctly that the values selected are related to the profile inclination (i.e., the slope)? I think the authors should investigate this in detail. Such value has no statistical-meaning regarding the correlation of the two geophysical parameters. Would not be more convenient to investigate the variograms of the measured data? Or, at least from the two independent inversions (following the smooth-constrained algorithm)? I might be misunderstanding this point, but the main inclination of the profile is not an argument to define the correlation lengths in this inversion. If the authors are really using the slope of the profile as a correlation length-scale, would not be expected then that this inversion provides practically the same inversion result than the smoothness-constraint? Finally, both approaches would be controlled by a lineal increase in the seismic velocities, which is in both cases forces to the plane defined by the surface geophones. 6) I also think that the authors should present information about the data-error. It would be convenient to see the pseudosection of the resistivity data, and maybe the travel times of the seismic measurements to assess the data quality. Maybe the small variations observed between different inversion algorithms result only by fitting the same data to the low error parameters defined by the authors. In this regard, I would be very interested to see more details of the normal-reciprocal analysis conducted by the authors. Just based on the principle of the error model, I would like to understand how can the authors solve for a relative error of 1% as mentioned in their manuscript. Such error is too low for the high resistivity solved

in the inversion. Such low relative error is not consistent with the description of the authors regarding the high contact resistances and the problems setting the measurements. Is the analyses of the data based on the misfit between normal and reciprocals or the fractional error? Which analysis was used to define the 300 ms error parameter defined in the inversion of the seismic data? I just find such values extremely low and would be critical to understand how were such values quantified. What were the steps used for the identification and removal of erroneous measurements and outliers? I think that the authors could then present the L-curve for their independent inversions (for such low error parameters) as this would make the study more complete. This could also alleviate concerns regarding the accuracy of the fitting in the inversion and remove the redundant Figures 5 and 8. 7) I think that the authors could improve the figures presented. I read the manuscript printed in hard copy and it was just impossible to read Figure 1 and the color bars (especially in Figure 2). It is clearly needed to read the digital file and zoom-in. Moreover, if the authors decide to keep the visual inspection/comparison of their results, it would be more convenient to have all results for one glacier plotted in a single figure (independent, joint petrophysical and joint structural inversion). Maybe the plot of the air/ice/water fraction resolved for both glaciers (after the joint-petrophysical inversion) could be plotted together. In this regard, it is possible to compare the resistivity and velocity models obtained by different inversions in a single figure and the retrieved parameters for both glaciers (regarding the discrimination between active and passive). I also do not understand the sense of Figure 5 and Figure 8, as the authors refer to the RMSE and chi-square obtained in the inversion and the values are acceptable. I am not sure which extra details we can obtain from the relative residuals. In this regard, (and although it was already mentioned above), maybe it is still more convenient for the authors to address the data quality and quantification of data-error in detail, as well as to investigate the actual statistical correlation between the data and the effect in the retrieved models for different petrophysical parameters than those presented in Table 3 and Table 4. Maybe the authors can use my suggestions. I hope that my comments help the authors to re-structure

and define the presentation of their results. I believe that an adequate analysis of the concerns mentioned above could also sharp the identity of the manuscript, right now it is not a case-study but it is also not a methodological paper. Maybe the investigation of the different aspects mentioned above could strength the relevance of the study as a case-study with an improved interpretation of the geophysical signatures through the combined used of different joint inversion strategies. Also, I recommend the authors to revise their conclusions and just list the main outcome of this study, avoiding speculations or points already addressed in previous studies.

References mentioned in this revision: • Glover, P.W., Hole, M.J. and Pous, J., 2000. A modified Archie's law for two conducting phases. Earth and Planetary Science Letters, 180(3-4), pp.369-383. • Glover, P., 2009. What is the cementation exponent? A new interpretation. The Leading Edge, 28(1), pp.82-85.

Please also note the supplement to this comment:
https://tc.copernicus.org/preprints/tc-2020-306/tc-2020-306-RC1-supplement.pdf

---

## Referee Comment (RC2) · Anonymous Referee #2 · 21 Dec 2020

**Summary**:
The paper presents electrical resistivity and seismic refraction measurements from two Andean rock glaciers and its objective of the paper is twofold: On the one hand, the authors present new field data acquired under challenging conditions at an active (El Jote) and an inactive rock glacier (El Ternero) and discuss their hydrogeological roles in semi-arid Chile. On the other hand, the authors present a comparison of individual common geophysical inversions of both data sets as well as recently developed structurally-coupled and petrophysically-coupled joint inversion approaches. Both ob-

jectives are certainly of interest to the cryospheric-geophysical community. The former perfectly fits the scope of The Cryosphere (TC). (The latter as well, but I feel that it currently requires a lot of geophysical prior knowledge and that further explanations and reasoning are necessary for the audience of TC as I will outline in my general comments below.)

The paper starts with an introduction on the importance of monitoring rock glaciers in a warming climate, the benefit of surface-based geophysical measurements and means to combine geophysical data sets in joint inversion approaches. Section 2 introduces the two study sites. Section 3 briefly describes the theory of Electrical Resistivity Tompgraphy (ERT) and Refraction Seismic Tomography (RST) and the two joint inversion approaches. This is followed by results (section 4), discussion (section 5) and conclusions and outlook (section 6). The work is illustrated with 9 figures, most of which are high-quality vector graphics. The paper is generally well written, but contains multiple linguistic oversights.

While I find the content of the paper very interesting and feel that the material compiled in this manuscript is generally well suited for publication (i.e., both the field data as well as the comparison of joint inversion approaches in a permafrost context are novel), the current manuscript requires major revisions. The authors are kindly requested to refer to my general, as well as line and figure-specific comments below during revision of their manuscript.

**General comments**:

**Target audience** I feel that the two objectives (both of which are really interesting) present a challenge, because the latter (comparison of structurally and petrophysically-coupled joint inversion approaches) requires a lot of prior knowledge on the two inversion approaches (and regularized inversion in general) considering that TC targets a broad (and not necessarily geophysical) audience. In

contrast, a reader with an expertise in geophysical (joint) inversion would probably be interested in a more detailed comparison of the two approaches, i.e., a comparison which allows to see under which circumstances one approach outperforms the other for instance. Such a comparison should also come with a discussion on the motivation of two approaches. For example: Are structurally-coupled joint inversions (and the underlying assumption of structural similarity) appropriate in a permafrost context, where a transition from ice to air can result in an order of magnitude change in velocity, while electrolytic conduction stays negligibly low?

**Scope and objectives** Somewhat related to the previous point, I question if the 3 inversion approaches and their comparison are actually necessary for the conclusions drawn in this paper. In the key figure 9 for example, which the authors use for drawing several conclusions and recommend as a diagnostic tool for future studies, only 2 of the 3 inversion approaches appear and the corresponding inverted velocity and resistivity distributions and thus also the scattered points look very similar. This makes me wonder, if this case study could be presented with individual inversions only, while a follow-up study could then focus on a detailed comparison of joint inversion approaches in a permafrost context.

**Brevity** Some figures are discussed too briefly. The first paragraph in subsection 4.1 for example ends with the sentence "The model results for El Jote are given in Fig. 2(a) and (b)." (line 275). This should be directly followed by a discussion on what can be seen in Fig. 2. The reader is left alone here until the figure is briefly mentioned again in the next subsection (4.2, line 322). Furthermore, subsection 4.1 ends with a single sentence on which quantity is plotted in Fig. 5. A further discussion on this figure and the shown residuals is missing.

**Structure:** The paper formally follows a standard structure, i.e. introduction, methods, results, discussion, conclusions and outlook. However, the current version of the

manuscript deviates from this structure several times, which is confusing for the reader. For example section 3 "Methods" contains a lot of theory and could be re-named more appropriately to "Theory and methods". Furthermore, many details with regard to the processing (e.g., used correlation lengths in the geostatistical regularization, choice of regularization strengths using L-curve analysis, choice of starting model, etc.) appear in the results section (rather than in methods). The manuscript would benefit from a clearer differentiation between theory, methods and results.

**Missing information / lack of clarity** I had problems following the data acquisition and processing. For example: Where were the off-line shots located? I feel that an additional figure illustrating the roll-along scheme and source/receiver positions, potentially in combination with Fig. 1, would come a long way here. With regard to the processing, not much information is given. How was the data quality? How did the authors process and filter the data sets? Please provide a plot with raw and filtered seismic and electrical data (e.g., apparent resistivities and apparent velocites) and explain the filtering steps applied.

**Specific comments**:

- First line of abstract: "four-times" → "four times"

- Third line of abstract: Please rephrase or explain "human pressure"

- L14: "four-times" → "four times"

- L45: "semiarid" was written with hyphen ("semi-arid") earlier in the abstract. Be consistent.

- L56: This sounds as if there were only two options, but borehole-based geo-physics and a combination of approaches exist as well.

- L74: "their" is unfortunate here as it could refer to both the glaciers and the inversion methodologies.

- L105: Space missing between number and unit (3020 m)

- L130: "Geophysics measurement" → "Geophysical measurements"

- L136: "collects" sounds a bit too easy (data is easily collected, but the parameter estimation is a bit more tricky). Maybe use "tries to infer" or "aims to estimate" instead.

- L152: "Following Maurer and Hauck (2007) methodology" → "Following the methodology of Maurer and Hauck (2007)"

- L210: Does it really enforce structural similarity? I think "promote" would be more correct here.

- L223: So what's the difference to the superposition of damping and smoothing then?

- Eq. 5: Is $\psi_m$ the same as in equation 1 here, i.e. does the petrophysical joint inversion use geostatistical regularization as well? Otherwise, I'll recommend to use a different symbol.

- L243: "constrain" → constraints"

- L255: "no-physical" → "non-physical"

- L259: Please provide more justification here. How and for which substrate types were the literature values determined? What assumptions are implied by using them to your study sites?

- L261: Remove comma after Wagner et al. (2019).

- L269: "we implemented". Was it really implemented from scratch or were codes available / provided to you? Please specify or reformulate to "we applied based on the implementations provided by Jordi et al. [...]".

- L271 and L319: I suggest to use $\rho_a^{\mathrm{median}}$ instead of $\rho a_{\mathrm{median}}$. Otherwise, the apparent resistivity $\rho_a$ cannot be differentiated from the resistivity times a factor $a$, i.e. $\rho a$.

- L276-279: These details would fit better in section 3.

- L293: "(Mollaret et al., 2020)" → "Mollaret et al. (2020)"

- L307: Subscripts are capitalized here (but not in equation 6).

- L309-311: This would be better suited in the section of data acquisition.

- L328: I appreciate that the authors use the same colorbar limits to allow visual comparison. I think this is valuable between the inversion approaches of a single profile, but there is no reason to keep it the same for the different sites as well. It is somewhat unfortunate that velocities of up to 7000 m/s appear for El Ternero, while the colorbar is limited to 4000 m/s. As a consequence, Fig. 6 is mainly yellow.

- L369-L370: Please elaborate: What is meant by remaining ambiguity within the interpretation of observations?

- L406: Redundant space after opening bracket

- L412: It looks like the colors in Fig. 9 have changed (red is joint inversion) and black vs. blue.

- L462: Missing space after period.

- L467: Missing space after comma.
* * *

---

## Referee Comment (RC3) · Lukas U. Arenson (Referee) · 28 Dec 2020

First, I'd like to apologize for the delay in reviewing the manuscript. It has been an extremely challenging year for everyone, I assume. De Pasquale and co-authors present a manuscript in which they compare 2D joint inversion routines of electrical resistivity and refraction seismic tomography (ERT / RST) from an active and an inactive rock glacier located in the Andes. Considering the challenging circumstances for collecting the data, the results of the geophysical survey are of interest to the readers of this Journal. However, the paper lacks a clear focus as the authors are

trying to cram too many ideas and thoughts, some supported by good evidence and others purely speculative. Therefore, I suggest that the manuscript should not be published in its current form and major revisions are required. As a supplement, the authors will find an annotated version that many contains specific comments and questions as well as editorial suggestions. My comments here are therefore only of general nature. I understand that the novel joint inversion is the key of the research and as such, the authors should focus on those measurements and results. The discussion on the hydrological significance is not essential for this publication and in fact distracting. In addition, there seems to be several misconceptions regarding the hydrogeology. For example, the authors imply that water in the watershed must originate from a cryoform. That is incorrect and I think the measurements seem to indicate that there are relatively shallow groundwater aquifers likely below the base of the rock glacier. A proper understanding of the hydrogeology would be needed prior to drawing the conclusions presented, but the authors do not provide any information on the hydrogeology. The measurements also do not support a discussion on the periglacial hydrology as presented, and it would really be better to completely delete these sections. I would also encourage the authors to carefully read some of the articles referenced so that they accurately cite these references and not just sentences that may be read out of a general context. After reading the manuscript I'm still confused about the El Jote Rock Glacier. Is it now an inactive rock glacier, or is it a relict rock glacier? Based on the inversion results it seems that the average (!) volumetric (I assume it is volumetric) ground ice content is 1 – 2%. Unfortunately, the authors did not provide any error ranges for their outputs (something that must be added in the revised version), but even if the error is +/- 5%, which would be very good, this landform is more likely to not contain any ground ice anymore. This means, the probability for the El Jote Rock Glacier being a relict rock glacier, i.e. there is no permafrost left, is significantly higher than it being an intact rock glacier (active or inactive). The new inversion presented seems reasonable, however, there is very little evidence for it to be accurate because there are no in-situ data available, as the

authors indicate. I'd like to remind the authors that geophysical investigations have been completed by others for which data from boreholes are available. The authors are therefore encouraged to first test their new approach for a well-known site and once confirmed that the methodology is accurately working, applying it to a site for which no information is available can be done. I was also surprised why the authors did not collect any soil samples from the front of the rock glaciers to at least get an idea of the potential gradation of the soil material and some of its characteristics, but instead they rely on references from the Alps. It also would have been helpful if the authors had extended their lines past the edge of the cryoforms and carried out additional lines perpendicular to the only one they completed, which would have allowed them to measure the ERT and RST characteristics of the natural terrain without a rock glacier as well as providing a cross calibration point. This is a fundamental step when completing geophysics on rock glaciers without any other information. Finally, I'm very surprised by the depth of the surveys. The authors managed to go much deeper than most ERT and RST surveys using similar configurations and I could not find an explanation for that. It is important that the authors better acknowledge the very limited data they have. It is understood that the measurements are challenging to complete, but this major limitation must be reflected in the interpretation of the results, the discussion and ultimately in the conclusions drawn from the two, very different surveys. Finally, there are several conceptual problems in the manuscript, such as when it comes to the origin of the water, or calling the form El Ternero glacier, instead of El Ternero rock glacier, saying that the rock glacier surface is below a layer of rocks, setting the permafrost table equal with the top of an ice-rich layer, or implying that an inactive rock glacier must be in a degrading state, etc.. While some of these mistakes may sound minor, they are indicative for not taking proper care of the science and rushing through arguments without taking care of every single sentence and word written. In summary, the measurements are worth to be published in the Cryosphere, but a major revision of the manuscript is strongly recommended for which the authors should focus on the novelty and refrain from speculations.
Please also note the supplement to this comment:
https://tc.copernicus.org/preprints/tc-2020-306/tc-2020-306-RC3-supplement.pdf
* * *
[Figure]

**Supplement:**

[revised manuscript text omitted]

$$f_\mathrm{r} + f_\mathrm{w} + f_\mathrm{i} + f_\mathrm{a} = 1. \qquad (2)$$

The treatment of the rock volumetric fraction as a single phase is a justified simplification in rock glacier environment, where the amount of soil is negligible compared to the hard rock.

240  The volumetric fractions in Eq. (2) are related to the seismic slowness (s), reciprocal of the P-wave propagation velocity (v), through the time averaging equation (Timur, 1968):

$$s = \frac{1}{v} = \frac{f_\mathrm{r}}{v_\mathrm{r}} + \frac{f_\mathrm{w}}{v_\mathrm{w}} + \frac{f_\mathrm{i}}{v_\mathrm{i}} + \frac{f_\mathrm{a}}{v_\mathrm{a}}, \qquad (3)$$

and to the electrical resistivity through a modification of Archie's second law (Archie, 1942):

$$\rho = \rho_\mathrm{w}(1 - f_\mathrm{r})^{-\mathrm{m}} \left( \frac{f_\mathrm{w}}{1 - f_\mathrm{r}} \right)^{-\mathrm{n}}, \qquad (4)$$

[revised manuscript text omitted]

---

## Author Comment (AC1) · 22 Jan 2021

Dear editor, With respect to the comments from Referee #1, we have addressed the major concerns indicated by the reviewer in this document and minor changes suggested in tc-2020-306-RC1-supplement.pdf file (attached). The attached files also include figures that have been modified in response to the comments from Referee #1,2 and 3.

1) I feel the title to be too provocative and/or to certain extent misleading. I understand

that the authors want to stress the comparison of results obtained through two different inversion algorithms. However, I am not sure that the comparison is valid taking into account that one refers to a spatial regularization schemes and the other one aims at solving a set of petrophysical models. Moreover, the joint-petrophysical inversion uses a set of calibration parameters (presented in Table 3 and Table 4) to permit the computation of air, ice and water content. However, in the present study, such parameters are taken from the literature, even if these are calibration values than need to be adjusted to each site (see the works from Archie and the reference to the studies by Glover et al., (2000) and Glover (2009)– full references below). In this regard, the comparison is unfair and technically limited. Would not be better to change the title to something like "extended interpretation based on the application of two joint-inversion algorithms"?

Reviewers 2 and 3 commented that the manuscript was lacking a clear focus, required a better structure and according to this review it does not sufficiently investigated the comparison between the two joint inversion results. In an attempt to respond to all of these comments we decided to change the focus of the manuscript to the geophysical signature difference between active and inactive rock glacier and have updated the title accordingly. The new proposed title is now "Geophysical signature of two contrasting Andean rock glacier". In the new version of the manuscript, we focus on the individual inversion results and present the petrophysical joint inversion to aid the interpretation of the differences in the geophysical signature of the two rock glaciers and completely delete the structural joint inversion approach.

2) If the authors decide that the comparison of the results is relevant, then I suggest that the authors provide a quantitative comparison of the parameters obtained through the joint inversion, i.e.,the seismic velocities and electrical resistivity resolved from the 2 inversion algorithms.Right now, the authors present only the plot of the inversion results and force the readers to compare those results by means of color-coded images in different pages and sizes. I feel such comparison to be at best qualitative and open to debate. It would be better if the authors plot the parameters solved for both strategies (for example, Vp from the joint-petrophysical inversion vs. Vp from the structural joint inversion). In this case, deviations between both approaches could be quantified. Moreover, such analysis would be also convenient within a numerical study (with Gaussian error), where deviations from the truth model can be also quantified.

We agree with the referee that such analysis is necessary for a clear comparison of the two joint inversion schemes but we feel that for this paper such a detailed quantitative comparison would make the paper too dense and we have opted to focus on the geophysical signature of the two rock glaciers as stated before. A possible solution as suggested by reviewer 2 would be a follow up paper with a more detailed analysis of the two inversion schemes, where we could focus as well on a numerical study to better quantify the accuracy of the two inversion schemes.

3) In order to perform a proper analysis of the two algorithms, the authors should investigate the variations in the retrieved models after testing different parameters used for the inversion. In this regard, the authors could investigate the resulting seismic velocities and electrical resistivity values after testing a few parameters in the petrophysical joint inversion and a few combinations of the scale-length correlations for the joint-structural inversion. Right now, the study runs a set of inversions with some values extracted from the literature (for the petrophysical inversion), and based on the slope (for the structural inversion. Are we expecting the models to be comparable? – Actually we are forcing the joint inversion to converge with some predefined settings that might not accurately describe the field conditions. In this regard, the users might be causing larger uncertainties in the inversion than just solving for a smooth-constraint independent inversion of the different data sets. I think that the proper comparison of the different joint-inversion algorithms needs to address the use of adequate parameters, or at least assess the deviations in the retrieved models by an inadequate selection of the inversion settings. The use of joint-inversion schemes has been largely investigated in geophysical studies, still they are not widely-accepted as they rely on the use of site-specific models or require of a correlation between the different parameters that

might not exists. I think the authors point out to this problem. The use of a numerical study would be also a good option to extend the analysis to quantify deviations from the truth model.

The revised version of the paper does not compare the two algorithms, so this reviewer comment is no longer relevant (see comments above).

4) Regarding the correlation of the seismic and electrical parameters, I find Figure 9 quite intriguing. Actually, the authors demonstrate no correlation between the seismic velocities and the electrical resistivity. I can see a cloud of points with a large variance and to pattern. However, the authors describe a correlation and quantify a model linking both of the properties. However, the authors do not present the correlation coefficients that actually quantify the actual correlation. The actual lack of correlation observed in Figure 9 is especially disturbing taking into account that the use of the joint-structural constraints aims at improving such correlation. Is such poor correlation due to the inadequate correlation lengths selected in this study? Is this a problem of poor data quality? If the authors cannot address this question in detail, I think that the authors should completely remove Figure 9. If the authors decide to keep the figure, please write explicitly the correlation coefficient and address in detail the lack of correlation.

For the comparison and analysis of the geophysical signature we have modified Fig.9 with a density plot of the resistivity and velocity model inversion parameters (now Fig.8). The corresponding section in the discussion has been changed as follows:

5.4 Towards a diagnostic model representation for the ice presence in rock glaciers. The results from the petrophysical joint inversion help quantify the volume content of air, water, ice and rock and identify El Jote as relict and El Ternero as intact rock glaciers. However, in many cases such an interpretation is limited by the lack of proper petrophysical models (or parameters). When petrophysical model coupling is not possible, the comparison of velocity and resistivity model inversion results can still deliver plenty of information about the rock-glacier's internal structure. In Fig.8 we show

resistivity-velocity density plots for each rock glacier, built from the individual model inversion results of figures 4(c),(d) and 6(c),(d). The differences between the two rock glaciers are clearly noticeable, with relatively low resistivity and low velocity clusters for the relict rock glacier, while the intact one is associated with higher velocities and resistivities. The relatively low resistivities and low velocities (Fig. 8a) are in agreement with air filled unconsolidated sediments inferred through the petrophysical joint inversion results (Figs. 5e,f). The lowest resistivities may be associated with liquid water and/or a proglacial aquifer (Fig. 5c; section 5.2). The gradual increase in resistivity and velocity (Fig. 8b) are evidence of material consolidation such as bedrock or ice-rich layers. Given the very high resistivities (over $10^5$ Ohm m) our interpretation is that these are ice rich layers (Table 1, resistivity values), which agrees with the petrophysical joint inversion results (Fig. 7d). The rather different appearance of the two density plots (Fig. 8a and b) can be used as an indicator of the distinct nature of the two rock glaciers: overall, the stagnant rock glacier is characterized by lower resistivities and velocities while the intact rock glacier is indicated by higher resistivity and velocity values, reflecting the ice rich layer. The schematic plot (Fig. 8c) summarizes the findings for our two end-member rock glaciers and could be useful for identifying ice-rich landforms using seismic and electrical resistivity methods.

5) I would like to get further information regarding the reasons to select the correlation-lengths used in the joint-structural inversion. Did I understood correctly that the values selected are related to the profile inclination (i.e., the slope)? I think the authors should investigate this in detail. Such value has no statistical-meaning regarding the correlation of the two geophysical parameters. Would not be more convenient to investigate the variograms of the measured data? Or, at least from the two independent inversions (following the smooth-constrained algorithm)? I might be misunderstanding this point, but the main inclination of the profile is not an argument to define the correlation lengths in this inversion.If the authors are really using the slope of the profile as a correlation length-scale,would not be expected then that this inversion provides practically the same inversion result than the smoothness-constraint? Finally, both approaches

would be controlled by a lineal increase in the seismic velocities, which is in both cases forces to the plane defined by the surface geophones.

We agree with the referee that a better study needs to be done in order to include the structural joint inversion; therefore we avoid presenting these results in the paper which are not relevant considering its new focus.

6) I also think that the authors should present information about the data-error. It would be convenient to see the pseudosection of the resistivity data, and maybe the travel times of the seismic measurements to assess the data quality. Maybe the small variations observed between different inversions algorithms result only by fitting the same data to the low error parameters defined by the authors. In this regard, I would be very interested to see more details of the normal reciprocal analysis conducted by the authors. Just based on the principle of the error model, I would like to understand how can the authors solve for a relative error of 1% as mentioned in their manuscript. Such error is too low for the high resistivity solved in the inversion. Such low relative error is not consistent with the description of the authors regarding the high contact resistances and the problems setting the measurements.Is the analyses of the data based on the misfit between normal and reciprocals or the fractional error? Which analysis was used to define the 300 ms error parameter defined in the inversion of the seismic data? I just find such values extremely low and would be critical to understand how were such values quantified. What were the steps used for the identification and removal of erroneous measurements and outliers?I think that the authors could then present the L-curve for their independent inversions (for such low error parameters) as this would make the study more complete. This could also alleviate concerns regarding the accuracy of the fitting in the inversion and remove the redundant Figures 5 and 8.

In responding to this comment we realized that the definition of the error model in the original manuscript was incorrect and it has since been modified. The errors were computed for ERT data using the mean standard deviation of the observations which was of 1.2 % in case of El Jote and 11.4 % in case of El Ternero. These were the

actual values used for the inversion schemes and we modified the text and table accordingly. For the error of seismic data we calculated an average error of 0.001 s as an estimate of the average variability in our picking of the first arrival traveltime. In the new proposed manuscript we also present a new section of the Methods: 3.3 Data processing and Inversion where. Here we clarify how the filtering happens: "The ERT observation were automatically filtered by the acquisition software which did not take measurements when the contact resistance was too high, while for the seismic refraction traveltime, we manually picked the first arrivals after applying a gain to the seismic traces and therefore the traces were filtered according to our ability of identify the first arrival times." Also, we eliminated Figure 5 and 8 and added a new Figure 3 with the L-curve analysis for the individual inversions. Moreover, we provided images of the datasets for both rock glaciers (first arrival traveltimes for RST and pseudosections for ERT ) in new Figures 4 and 5 where we presents the individual inversion results.

7) I think that the authors could improve the figures presented. I read the manuscript printed in hard copy and it was just impossible to read Figure 1 and the colour bars (especially in Figure 2). It is clearly needed to read the digital file and zoom-in. Moreover, if the authors decide to keep the visual inspection/comparison of their results, it would be more convenient to have all results for one glacier plotted in a single figure (independent, joint petrophysical an djoint structural inversion). Maybe the plot of the air/ice/water fraction resolved for both glaciers (after the joint-petrophysical inversion) could be plotted together. In this regard,it is possible to compare the resistivity and velocity models obtained by different inversions in a single figure and the retrieved parameters for both glaciers (regarding the discrimination between active and passive). I also do not understand the sense of Figure 5 and Figure 8, as the authors refer to the RMSE and chi-square obtained in the inversion and the values are acceptable. I am not sure which extra details we can obtain from the relative residuals. In this regard, (and although it was already mentioned above), maybe it is still more convenient for the authors to address the data quality and quantification of data-error in detail, as well as to investigate the actual statistical correlation between the data and the effect in the

retrieved models for different petrophysical parameters than those presented in Table 3 and Table 4.

We have improved the quality of the figures as suggested (increasing the font size of the labels and dividing Figure 1 in two new figures 1 and 2). Given the new scope of the paper we eliminated Figures 5 and 8.

Also we would like to specify that it was not possible to upload the figure with the complete caption because of their lengths. Please find below the complete caption of all figures:

Figure 1. (a) Overview map indicating the location of Estero Derecho ($\sim 30°$S, $70°$W) in the Coquimbo Region of Chile. Elevation map from ASTER GDEM. (b) Detailed map of Estero Derecho with an inventory of landforms created by CEAZA. The delineations for El Jote and El Ternero were created specifically for this study from the Esri base-map satellite imagery. Both landforms are labeled with their respective elevation ranges.

Figure 2. (a) Aerial image of El Jote, showing the location of the geophysical survey line and (b) its topography from field differential GPS measurements. (c) Aerial image of El Ternero, showing the location of the geophysical survey line and (d) its topography from field differential GPS measurements. Base maps in (a) and (c) from Esri World Imagery 2018. (e) Scheme of the 50 % roll-along scheme used for ERT surveys on both rock glaciers and RST survey on El Jote. (f) Scheme of geophones and Inline/Offline shot positions for RST surveys.

Figure 3. L-curve analysis for the regularization weights (Lambda) used in the inversion of ERT and RST data on both rock glaciers. In each plot, the values tested are Lambda= 1, 5, 10, 15, 50, 100.

Figure 4. Geophysical observations and inversion model results for El Jote rock glacier. (a) RST first arrival traveltimes. (b) ERT apparent resistivity. (c) Velocity and (d) resistivity tomograms. The velocity model is cut below the lowermost ray-path while the

resistivity model transparency is proportional to the ERT data coverage. The velocity colorbar is linear, while the resistivity one is expressed in logarithmic scale.

Figure 5. Petrophysical joint inversion results of El Jote field data sets. The tomograms represents (a) velocity and (b) resistivity transformed models. The directly inverted parameters are (c) water, (d) ice, (e) air and (f) rock volumetric content. All models are cut off below the lowermost ray path, with only resistivity colorbar expressed in logarithmic scale.

Figure 6. Geophysical observations and inversion model results for El Ternero rock glacier. (a) RST first arrival traveltimes. (b) ERT apparent resistivity. (c) Velocity and (d) resistivity tomograms. The velocity model is cut below the lowermost ray-path while the resistivity model transparency is proportional to the ERT data coverage. The velocity colorbar is linear, while the resistivity one is expressed in logarithmic scale.

Figure 7. Petrophysical joint inversion results of El Ternero field data sets. The tomograms represents (a) velocity and (b) resistivity transformed models. The directly inverted parameters are (c) water, (d) ice, (e) air and (f) rock volumetric content. All models are cut off below the lowermost ray path, with only resistivity colorbar expressed in logarithmic scale.

Figure 8. Density plots of resistivity versus P-waves velocity values for (a) El Jote and (b) El Ternero datasets. (c) Schematic plot of the qualitative ERT and RST signature for intact and stagnant rock glaciers.

Please also note the supplement to this comment:
https://tc.copernicus.org/preprints/tc-2020-306/tc-2020-306-AC1-supplement.pdf
* * *
[Figure]

**Fig. 1.** Overview map indicating the location of Estero Derecho, with (a)elevation map and (b)inventory of landforms ($\sim$ 30°S, 70°W) in the Coquimbo Region of Chile

[Figure]

**(a) El Jote**

**(b)**

**(c) El Ternero**

**(d)**

**(e)**
24-channels ERT/RST cable

50% Roll-along scheme

**(f)**
Offline shot

Inline shot

Geophone

10 m

5 m

**Fig. 2.** Aereal images of El Jote and El Ternero with field layout schemes

[Figure]

RST                    ERT
EL JOTE

model norm

EL TERNERO

residual norm          ● λ=10

[revised manuscript text omitted]

---

## Author Comment (AC2) · 22 Jan 2021

Dear editor, With respect to the comments from Referee #2, we have addressed the major concerns and specific comments indicated by the reviewer in this document. The figures modified accordingly to the comments of Referee#2 are included in the answer to Referee #1.

1) General comments Target audience: I feel that the two objectives (both of which are really interesting) present a challenge, because the latter (comparison of structurally

and petrophysically-coupled joint inversion approaches) requires a lot of prior knowledge on the two inversion approaches (and regularized inversion in general) considering that TC targets a broad (and not necessarily geophysical) audience. In contrast, a reader with an expertise in geophysical (joint) inversion would probably be interested in a more detailed comparison of the two approaches, i.e., a comparison which allows to see under which circumstances one approach outperforms the other for instance. Such a comparison should also come with a discussion on the motivation of two approaches. For example: Are structurally coupled joint inversions (and the underlying assumption of structural similarity) appropriate in a permafrost context, where a transition from ice to air can result in an order of magnitude change in velocity, while electrolytic conduction stays negligibly low?

Indeed, we agree with this review and we decided to change the title and focus of the manuscript to the geophysical signature difference between stagnant and inactive rock glaciers. The new proposed title is now "Geophysical signature of an intact and stagnant Andean rock glacier". In the new manuscript, we focus on the individual inversion results and present the petrophysical joint inversion to aid the interpretation of the differences in the geophysical signature of the two rock glaciers and completely delete the structural joint inversion approach.

Scope and objectives: Somewhat related to the previous point, I question if the 3 inversion approaches and their comparison are actually necessary for the conclusions drawn in this paper. In the key figure 9 for example, which the authors use for drawing several conclusions and recommend as a diagnostic tool for future studies, only 2 of the 3 inversion approaches appear and the corresponding inverted velocity and resistivity distributions and thus also the scattered points look very similar. This makes me wonder, if this case study could be presented with individual inversions only, while a follow-up study could then focus on a detailed comparison of joint inversion approaches in a permafrost context.

Accordingly to this and the previous comments, we re-structured and re-focused the

manuscript as briefly presented above. Figure 9 has now been changed with a resistivity-velocity density plot to address the lack of correlation pointed out by reviewers 1 and 3). The new figure is still useful as a diagnostic plot and we have rewritten section 5.4 accordingly:

5.4 Towards a diagnostic model representation for the ice presence in rock glaciers.

The results from the petrophysical joint inversion help quantify the volume content of air, water, ice and rock and identify El Jote as relict and El Ternero as intact rock glaciers. However, in many cases such an interpretation is limited by the lack of proper petrophysical models (or parameters). When petrophysical model coupling is not possible, the comparison of velocity and resistivity model inversion results can still deliver plenty of information about the rock-glacier's internal structure. In Fig.8 we show resistivity-velocity density plots for each rock glacier, built from the individual model inversion results of figures 4(c),(d) and 6(c),(d). The differences between the two rock glaciers are clearly noticeable, with relatively low resistivity and low velocity clusters for the relict rock glacier, while the intact one is associated with higher velocities and resistivities. The relatively low resistivities and low velocities (Fig. 8a) are in agreement with air filled unconsolidated sediments inferred through the petrophysical joint inversion results (Figs. 5e,f). The lowest resistivities may be associated with liquid water and/or a proglacial aquifer (Fig. 5c; section 5.2). The gradual increase in resistivity and velocity (Fig. 8b) are evidence of material consolidation such as bedrock or ice-rich layers. Given the very high resistivities (over 10ˆ5 Ohm m) our interpretation is that these are ice rich layers (Table 1, resistivity values), which agrees with the petrophysical joint inversion results (Fig. 7d). The rather different appearance of the two density plots (Fig. 8a and b) can be used as an indicator of the distinct nature of the two rock glaciers: overall, the stagnant rock glacier is characterized by lower resistivities and velocities while the intact rock glacier is indicated by higher resistivity and velocity values, reflecting the ice rich layer. The schematic plot (Fig. 8c) summarizes the findings for our two end-member rock glaciers and could be useful for identifying ice-rich landforms using

seismic and electrical resistivity methods.

Brevity: Some figures are discussed too briefly. The first paragraph in subsection 4.1 for example ends with the sentence "The model results for El Jote are given in Fig. 2(a) and (b)." (line 275). This should be directly followed by a discussion on what can be seen in Fig. 2. The reader is left alone here until the figure is briefly mentioned again in the next subsection (4.2, line 322). Furthermore, subsection 4.1 ends with a single sentence on which quantity is plotted in Fig. 5. A further discussion on this figure and the shown residuals is missing.

We address the brevity issue within the manuscript adding a proper description of each figure within the text. Also, because the focus is now on the individual inversion results as principal tool for the building of a diagnostic model, we presents the collected datasets and inversion results separately for each rock glacier in new Figure 4 (for El Jote) and Figure 6 (for El Ternero) and comment on those within the text. Moreover, we avoid presenting Figures 5 and 8 simply referring to the chi2 values as a measure of the goodness of fit.

Structure: The paper formally follows a standard structure, i.e. introduction, methods, results, discussion, conclusions and outlook. However, the current version of the manuscript deviates from this structure several times, which is confusing for the reader. For example section 3 "Methods" contains a lot of theory and could be renamed more appropriately to "Theory and methods". Furthermore, many details with regard to the processing (e.g., used correlation lengths in the geostatistical regularization, choice of regularization strengths using L-curve analysis, choice of starting model, etc.) appear in the results section (rather than in methods). The manuscript would benefit from a clearer differentiation between theory, methods and results.

We have followed the suggestion and changed the title of section 3 in Theory and Methods and summarize the processing and inversion parameter choices in a new subsection of it:

3.3 Data processing and Inversion The ERT observation were automatically filtered by the acquisition software which did not take measurements when the contact resistance was too high, while for the seismic refraction traveltime, we manually picked the first arrivals after applying a gain to the seismic traces and therefore the traces were filtered according to our ability of identify the first arrival times. The inversion algorithms we use in order to interpret the geophysical observations are part of pyGIMLI, an open-source library developed in python for geophysical inversion and modelling (Rücker et al., 2017). On each rock glacier we implement the same discretization mesh for both ERT and RST inversion routines and use a regularization weight of lambda= 10 for the inversion of all the dataset, chosen according to the L-curve analysis (Hansen, 2001). A schematic plot of the L-curve analysis for each dataset collected is given in Figure 3, in all cases we present the model solution L2-norm against the residual L2-normobtained for lambda=1, 5, 10, 15, 50 and 100. For both rock glaciers we use an homogeneous resistivity starting model, with a value equal to the median of the apparent resistivities (rho_mediana= 4561 Ohm m for El Jote and rho_mediana= 36054 Ohm m for El Ternero) and a gradient model for the seismic velocity, starting with 300m sˆ(−1) at the top of the tomogram and gradually increasing to 5000 m sˆ(−1) at the bottom of it. In each case, we refer to the error-weighted chi-square fit, where chi2= 1 signifies a perfect fit (Günther et al., 2006), to quantify the resulting model parameters' ability to explain the field observations. Moreover, in order to quantify the volumetric percentage of water, ice, air and rock within each of the two rock glacier, we implement the four phase petrophysical joint inversion scheme presented by Wagner et al. (2019) and tested in Mollaret et al.(2020). For this inversion scheme we kept the same discretization meshes used for the individual inversions. The methodological details regarding this inversion algorithm and its application for this case study are given in Appendix A.

Missing information / lack of clarity I had problems following the data acquisition and processing. For example: Where were the off-line shots located? I feel that an additional figure illustrating the roll-along scheme and source/receiver positions,potentially

in combination with Fig. 1, would come a long way here. With regard to the processing, not much information is given. How was the data quality? How did the authors process and filter the data sets? Please provide a plot with raw and filtered seismic and electrical data (e.g., apparent resistivities and apparent velocities) and explain the filtering steps applied.

We have addressed this point adding the new subsection 3.3 Data Processing and Inversion and clarifying in section 3.2 Acquisition strategy the location of the off-line shots on El Ternero: "The total length of 575 m was then obtained by moving the 24-channels set-up four times and adding off-line shots (Fig. 2f) to link the different acquisitions at distances of 5, 15 and 25 m from the last geophone at each end of the cable."

We have modified Figure 2, where we show aerial images of the two rock glaciers, to include the location of the geophysical survey line and its topography (from DGPS measurements) with schematics to clarify the roll along schemes and geophone-inline/offline shots positions. The filtering is addressed in section 3.3 Data processing and Inversion (above), and we show the filtered data in the new Figure 4 and 6.

2) Specific comments:

First line of abstract: "four-times"->"four times"

Modified accordingly in the manuscript.

Third line of abstract: Please rephrase or explain "human pressure"

Rephrased: human pressure on water resources.

L14: "four-times" -> "four times"

Modified accordingly in the manuscript

L45: "semiarid" was written with hyphen ("semi-arid") earlier in the abstract. Be consistent.

Modified as semi-arid.

L56: This sounds as if there were only two options, but borehole-based geophysics and a combination of approaches exist as well.

Reformulated: Rock glacier composition can be derived from direct observations (e.g., boreholes logs, outcrops, tunnels and temperature measurements), borehole and surface-based geophysical observations (Hausmann et al., 2007).

L74: "their" is unfortunate here as it could refer to both the glaciers and the inversion methodologies.

Reformulated: These methods ability. . .

L105: Space missing between number and unit (3020 m)

Modified accordingly in the manuscript.

L130: "Geophysics measurement" ->"Geophysical measurements"

Modified accordingly in the manuscript.

L136: "collects" sounds a bit too easy (data is easily collected, but the parameter estimation is a bit trickier). Maybe use "tries to infer" or "aims to estimate"instead.

Reformulated: ERT aims to estimate. . .

L152: "Following Maurer and Hauck (2007) methodology" -> "Following the methodology of Maurer and Hauck (2007)"

Modified accordingly in the manuscript.

L210: Does it really enforce structural similarity? I think "promote" would be more correct here.

The section 3.3.1 Structural joint inversion has now been removed from the manuscript.

L223: So what's the difference to the superposition of damping and smoothing then?

The section 3.3.1 Structural joint inversion has now been removed from the manuscript. Nevertheless damping in this case referred to a regularization which favors model close to a reference one while for smoothing it favors models with less parameter variability.

Eq. 5: Is Phim the same as in equation 1 here, i.e. does the petrophysical joint inversion use geostatistical regularization as well? Otherwise, I'll recommend using a different symbol. The section 3.3.1 Structural joint inversion has now been deleted and section 3.3.2 Petrophysical joint inversion is now in appendix. The regularization in case of petrophysical joint inversion is now the only one referred with the symbol ÏŢm

L243: "constrain" -> "constraints"

Modified accordingly in the manuscript.

L255: "no-physical" -> "non-physical"

Modified accordingly in the manuscript.

L259: Please provide more justification here. How and for which substrate types were the literature values determined? What assumptions are implied by using them to your study sites? Added in text: Such parameters are of value in periglacial environments and consistent with the relevant physical parameters for ERT and RST value also presented in Table 1, nevertheless geotechnical in situ measurements could improve the estimation of those and therefore the accuracy of the inversion model results.

L261: Remove comma after Wagner et al. (2019).

Modified accordingly in the manuscript.

L269: "we implemented". Was it really implemented from scratch or were codes available / provided to you? Please specify or reformulate to "we applied based on the implementations provided by Jordi et al. [...]".

This part has been removed in the new manuscript.

L271 and L319: I suggest to use rho_mediana instead of rho_aˆmedian. Otherwise, the apparent resistivity rho_aˆmedian cannot be differentiated from the resistivity times a factor a,i.e. rho_a.

Modified accordingly in the manuscript.

L276-279: These details would fit better in section 3.

This part has been removed from the new manuscript.

L293: "(Mollaret et al., 2020)" -> "Mollaret et al. (2020)"

Modified accordingly in the manuscript.

L307: Subscripts are capitalized here (but not in equation 6).

This equation has been removed from the text, agreeing with reviewer 1 we decided that such analysis was redundant and rely on chi2 for the quantification of the goodness of fit.

L309-311: This would be better suited in the section of data acquisition.

Modified accordingly in the manuscript.

L328: I appreciate that the authors use the same colorbar limits to allow visual comparison. I think this is valuable between the inversion approaches of a single profile, but there is no reason to keep it the same for the different sites as well. It is somewhat unfortunate that velocities of up to 7000 m/s appear for El Ternero,while the colorbar is limited to 4000 m/s. As a consequence, Fig. 6 is mainly yellow.

New figures presented with modified colorbar.

L369-L370: Please elaborate: What is meant by remaining ambiguity within the interpretation of observations?

This part has been removed from the manuscript.

L406: Redundant space after opening bracket

Removed in the manuscript.

L412: It looks like the colors in Fig. 9 have changed (red is joint inversion) and black vs. blue.

Figure 9 has been modified (see general comments)

L462: Missing space after period.

Modified accordingly in the manuscript.

L467: Missing space after comma.

Modified accordingly in the manuscript.

---

## Author Comment (AC3) · 22 Jan 2021

Dear editor, With respect to the comments from Referee #3, we have addressed the major concerns indicated by the reviewer in this document and minor changes suggested in tc-2020-306-RC3-supplement.pdf file (attached). The figures modified accordingly to the comments of Referee#3 are included in the answer to Referee #1. The paper lacks a clear focus as the authors are trying to cram too many ideas and thoughts, some supported by good evidence and others purely speculative.

[Figure]

With the aim of strengthening the focus of the paper we decided to follow the suggestion of Reviewer 2 and change the title and focus of the manuscript to the geophysical signature difference between active and stagnant rock glaciers. The new proposed title is now "Geophysical signature of two contrasting Andean rock glaciers". In the new manuscript, we focus on the individual inversion results and present the petrophysical joint inversion to aid the interpretation of the differences in the geophysical signature of the two rock glaciers and completely delete the structural joint inversion approach.

I understand that the novel joint inversion is the key of the research and as such, the authors should focus on those measurements and results.

The joint inversion analysis as presented in this paper is not novel. A comparable structural joint inversion analysis was presented in the paper by Jordi et al., 2019; and the petrophysical joint inversion was first presented in Wagner et al., 2019 and then tested for different parameters and sites in the study by Mollaret et al., 2020. As stated before, and following the suggestion of reviewer 2, we decided to focus on the geophysical signature rather than on the joint inversion methods completely deleting the structural joint inversion from the manuscript and leaving the petrophysical joint inversion as an interpretative aid to the individual inversion results.

The discussion on the hydrological significance is not essential for this publication and in fact distracting. In addition, there seems to be several misconceptions regarding the hydrogeology. For example, the authors imply that water in the watershed must originate from a cryoform. That is incorrect and I think the measurements seem to indicate that there are relatively shallow groundwater aquifers likely below the base of the rock glacier. The measurements also do not support a discussion on the periglacial hydrology as presented, and it would really be better to completely delete these sections.

Thank you for this critical review of the paper with regards to the hydrology. We agree that water may originate from groundwater sources as well as from the rock glacier. We have modified the sentence starting on line 310 to clarify our uncertainty in the interpretation and remove the part of the statement implying that the water originates from the rock glacier. We agree that the discussion on periglacial hydrology is tangential to the main results and conclusions of the paper. We have therefore significantly reduced section 5.5, maintaining those portions of the text that are most directly supported by the geophysical data. We do think it is valuable to maintain this section as it synthesizes some of the main geophysical results from El Jote and El Ternero and provides context for the importance of the findings with respect to their hydrological role.

 c After reading the manuscript I'm still confused about the El Jote Rock Glacier. Is it now an inactive rock glacier, or is it a relict rock glacier? Based on the inversion results it seems that the average (!) volumetric (I assume it is volumetric) ground ice content is 1 – 2%. (IS 1%) Unfortunately, the authors did not provide any error ranges for their outputs (something that mustbe added in the revised version), but even if the error is +/- 5%, which would be very good, this landform is more likely to not contain any ground ice anymore. This means,the probability for the El Jote Rock Glacier being a relict rock glacier, i.e. there is no permafrost left, is significantly higher than it being an intact rock glacier (active or inactive).

After careful review of the geophysical results we agree that El Jote should be classified as a relict rock glacier. We have modified the text to refer to this glacier as "stagnant" before the results are presented, and "relict" in the discussion section once the rock glacier has been interpreted as relict. Also, we quantified the maximum average volume content for different scenarios varying the initial porosity and porosity ranges within the petrophysical inversion results. These sensitivity results have been added to the paper and can be used to quantify the model error.

The new inversion presented seems reasonable; however, there is very little evidence for it to be accurate because there are no in-situ data available, as the authors indicate. I'd like to remind the authors that geophysical investigations have been completed by others for which data from boreholes are available. The authors are therefore encouraged to first test their new approach for a well-known site and once confirmed that

the methodology is accurately working, applying it to a site for which no information is available can be done.

As specified above, the petrophysical joint inversion presented in this paper is not novel and has been presented in Wagner et al.(2019) and thoroughly tested by Mollaret et al.(2020). It is true that the paper could be significantly improved with in-situ sampling data to validate the choice of the petrophysical parameters, but it was impossible to collect core data for this field study given the remote location and equipment available. For this reason, in the new manuscript the inversion approach has been moved to the appendix and used solely to aid in the interpretation of the individual inversion results.

I was also surprised why the authors did not collect any soil samples from the front of the rock glaciers to at least get an idea of the potential gradation of the soil material and some of its characteristics, but instead they rely on references from the Alps. It also would have been helpful if the authors had extended their lines past the edge of the cryoforms and carried out additional lines perpendicular to the only one they completed, which would have allowed them to measure the ERT and RST characteristics of the natural terrain without a rock glacier as well as providing a cross calibration point.

Thank you for this comment, we will consider collecting such soil samples in a future field campaign. During the field work for this study there logistical constraints impeded the complete geotechnical characterization of the material at the rock glacier front. Regarding the geophysical line we add the following lines to the manuscript:

"While the geophysical line extended slightly past the edge of the El Jote rock glacier, it was impossible to do so for El Ternero due to the high, steep, unstable and therefore dangerous slopes of the rock glacier front and lateral margins."

Finally, I'm very surprised by the depth of the surveys. The authors managed to go much deeper than most ERT and RST surveys using similar configurations and I could not find an explanation for that. It is important that the authors better acknowledge the very limited data they have. It is understood that the measurements are challenging

to complete, but this major limitation must be reflected in the interpretation of the results,the discussion and ultimately in the conclusions drawn from the two, very different surveys.

In the new version of the manuscript we added a new section in the discussion where we address the data quality:

5.1 Data quality and comparison of inversion routines

For both field sites the acquisition of data and their quality were limited by the environment: the presence of large boulders with air-filled voids between them at the surface of both glaciers attenuated both mechanical and electrical energy propagation.The quality of the data was especially affected in the case of El Ternero rock glacier, clearly demonstrated in Figures 4(a)-(b) and 6(a)-(b). It must be stressed that the parameter domains shown in the individual P-wave velocity inversion results and in the petrophysical joint inversion results (Figs. 4c, 6c, 5 and 7) are geometrically delimited by the lowermost ray path but the ray-coverage within the displayed area is limited. . ..

Finally, there are several conceptual problems in the manuscript, such as when it comes to the origin of the water, or calling the form El Ternero glacier, instead of El Ternero rock glacier, saying that the rock glacier surface is below a layer of rocks, setting the permafrost table equal with the top of an ice-rich layer, or implying that an inactive rock glacier must be in a degrading state, etc..

We are really thankful to the reviewer for their comments and have tried to correct and address them within the new manuscript.

Please also note the supplement to this comment:
https://tc.copernicus.org/preprints/tc-2020-306/tc-2020-306-AC3-supplement.pdf

**Supplement:**

[revised manuscript text omitted]

---

## Referee Report (RR1)

[referee-annotated manuscript omitted]

---

## Referee Report (RR2)

| n | 2.4 | $v_\mathrm{i}$ | 3500 (m s$^{-1}$) |
| m | 1.4 | $v_\mathrm{a}$ | 330 (m s$^{-1}$) |
| | | $v_\mathrm{r}$ | 6000 (m s$^{-1}$) |

795

[referee-annotated manuscript omitted]

---

## Referee Report (RR3)

[referee-annotated manuscript omitted]

---

## Author Response (AR2)

The anonymous referee #1 pointed out different concerns about the paper that we try to address as follows:

- In the current version of the manuscript, the results are presented and interpreted without addressing any concern regarding the uncertainty inherent to the inversion, for instance the non-uniqueness or the influence of poor data into the final inverted model. The lack of any ground truth or complementary information hinders any discussion of the inversion results. Likewise, the interpretation of the results is rather speculative due to the lack of ground truth and the limited amount of geophysical data.

*We now address the inversion uncertainty due to poor data quality (especially within Ternero rock glacier) within the discussion and stress that the use of different inversion routines and therefore different prior assumptions on the inversion problem help distinguish between features that are constrained by the data and those that are present as a result of the assumptions implicitly or explicitly imposed on the solution. The coherency between structures of relative high/low velocity and high/low resistivity within the inversion model results from two different schemes (proved by the moderate to good correlation found within the modeled velocities and resistivities), indicates that our data can constrain such features, therefore we are confident that this output can be used as base for the interpretation.*

- I recommended the conduction of numerical investigations to enhance the selection of parameters for the inversion schemes presented before. The authors decided to avoid that and just present the qualitative interpretation of the inverted images.

*We have now added a paragraph within the appendix where we discuss the choice of parameters for the petrophysical joint inversion and their effect within the inversion model results in more detail. Also, we added a new figure (Fig 8) to address the comparison of the inversion results quantitatively and not only through the comparison of the model images.*

- In my opinion, the presentation of inversion results of two disconnected profiles does not constitute a case study…. However, in my opinion, the two profiles are not enough to understand the hydrogeological dynamics of the site, and the discussion provided does not provide any news insights on the application of geophysical methods for the characterization of rock glaciers.

*We agree with the reviewer that the data are limited (and stress it within the discussion) but as addressed in a previous comment, we feel that the comparison of different inversion results and the coherency within the inversion model structure found is enough for the interpretation proposed within the paper.*

- The manuscript shows a clear lack of balance. There is lengthy introduction, where different hydrogeological aspects are discussed. However, there is no real incorporation of

the geophysical data into hydrogeological model. I am not sure whether the scarce geophysical information could be of any help in the groundwater management of the site. Maybe an alternative is to present further data aiming at quantifying the total ice content in one (or each) of the rock glaciers? – The presentation of two or three profiles collected over the area may also permit to evaluate the uncertainty of the inversion results, for instance comparing the values resolved for the same location using data collected in perpendicular profiles.

*We tried to slightly modify the introduction adding some relevant studies and shorten some f the paragraphs but feel the information included is all relevant for the study. Moreover, it was not possible (due to lack of resources) to add new data to the study but we hope the new numerical analysis presented in paper and the deeper discussion about the data and inversion process are sufficient.*

- Maybe also the authors could present a more careful analysis of the different parameters used within the inversion, for instance based on the discussion of Mollaret et al. (2020) regarding different regularization parameters, Archie's constants, or including even a poor estimation of surface conductivity.

*Within the appendix we present a paragraph where we discuss the choice and effect of different parameters (regularization weights, Archie's parameters and porosity ranges and initial values) and added a short discussion about the method limitations where we address the choice of resistivity in the petrophysical model (referring to Mollaret etal. 2020, the introduction of surface conduction terms within the resistivity petrophysical model doesn't modify the results since the parameters of the different empirical laws are similarly determined by minimizing the data misfit).*

- I also think that the authors could provide more information regarding the quality of their data, especially the seismic data presented in Figure 6 show ray paths that evidence a poor processing of the data. Are these trajectories really reproduced in the inversion? Maybe this is the reason that the inversion does not converge. In this regard, I would recommend the authors to provide a more careful analysis of the data. It is not adequate to interpret results with no convergence and a chi-square value of two is clearly an evidence of the inversion not being able to explain the data, even with the relative high error parameters mentioned by the authors. This example clearly demonstrates that there is a big issue with the data presented, thus, with the inverted images. However, such issue is never addressed in the manuscript and the hydrogeological parameters are interpreted as valid.

*Indeed, regarding Figure 6a, the ray paths look noisier than the first rock glacier because the temporal scale on the Y-axis is more limited than in Fig 4a (in the new Figure 6a, we now use the same Y-axis as for Figure 4). Moreover, the picks show indeed more high frequency variability, but this is due also to high frequency altitude variation within the big boulders of this rock glacier. In fact, the surface of this rock glacier is very chaotic with strong altitude variation between*

*geophones and sources positions. This effect generates high frequency picks lateral variation. Nevertheless, those picks are well modeled by the inversion process because of the quite low X² value (0.93 with an error of 1ms). So, we are confident about our data and model quality.*

*Regarding the ERT data for El Ternero, we increased the error from 10% to 15% and the X² is now 1,49 for the new individual inversion results. We modified the petrophysical joint inversion accordingly. In the corresponding figures we plot the new results. We also present the readings of currents and voltages for both El Ternero and El Jote ERT profiles below so the data quality may be reviewed.*

- **Ternero intact Rock Glacier**

*Voltage readings and injected current, before spatial filtering of C and P effects. The graphs display voltages from 4 to 160mV with an injected current from 0.05 to 0.5 mA and have been filtered to only include data with a standard deviation inferior to 25% and with an average of 2.5%.*

*1ˢᵗ line with Roll-along*
*(Voltage readings in mV and current injected in mA)*

[Figure]

*2ⁿᵈ line with Roll-along*

[Figure]

[Figure]

**3rd line with Roll-along**

[Figure]

**4th line with Roll-along**

[Figure]

- ***Jote relict Rock Glacier***

*Voltage readings and injected current, before spatial filtering of C and P effects. The graphs display voltages from 12 to 100mV with an injected current from 0.3 to 35 mA and have been filtered to only include data with a standard deviation inferior to 25% and with an average of 0.6%.*

[Figure]

**2$^{nd}$ line with Roll-along**

[Figure]

**3rd line with Roll-along**

[Figure]

**4th line with Roll-along**

[Figure]

**5th line with Roll-along**

[Figure]

[Figure]

Data and quadripoles configuration

- I am also attaching a marked PDF ( for spelling and grammatical errors), which may be of use for the authors to revise sections of their manuscript, in case the study is accepted by the recommendation of the other reviewers, or if they decide to submit it to another journal.

*We address the comments in theattached pdf: tc-2020-306-referee-report#1*

**Anonymous REFEREE #3**

- Rock. The manuscript uses "rock" synonym to solid particle. This is incorrect and leads to mistakes such as consolidated rock material (Line 320). In that line (320) the manuscript also talks about consolidated ice-rich material, which again is not correct in the absence of associated consolidation pressure.

*We clarify the formulation within the text but we left rock as terminology for the petrophysical joint inversion for coherence with the original formulation of the four phase model, which assumes that the contribution of sediments is negligible compared to hard rock.*

- The authors also need to pay attention on how the porosity is defined. Phase change doesn't affect porosity, unless you have an increase in the pore volume in response to the expansion of the ice during freezing.

*We address this concern following the comments within the attached pdf tc-2020-306-referee-report#3.*

- The authors claim that the acquisition of data and their quality were limited because of the environment. This is not accurate and the fact that the group had limit resources can't be blamed on the environment. If the group had spent another 4 weeks and had better equipment, it would have been different.

*We modify the sentence accordingly within the text.*

- Throughout the manuscript the authors use the term reserve (or reservoir) as well as resource. However, these terms have not been defined and since these definitions are not

clear it would be good if the authors were to define what it means in the context of the research.

*We modify the terminology accordingly to the comments given within the pdf tc-2020-306-referee-report#3.*

- At various places in the manuscript it would be good if the authors were to define if they are using volumetric or gravimetric percentages. There is a significant difference.

*We defined this at Line 250: "…these results with quantification of the volumetric content of the different subsurface components…"*

*Moreover, we now clarify this in the legend of figures 5 and 7.*

- Line 35: "which suggest that rock glaciers in the European Alps have experienced increased melt rates in recent decades". However, a rock glacier cannot melt. Melting is the physical process of phase change from solid to liquid. This is a scientific publication, and the authors must use precise terminology.

*Modified accordingly within the text: "…which suggest that rock glaciers in the European Alps have experienced increased ground ice melt and permafrost degradation rates in recent decades…"*

- Line 127: "…even though their employment in mountainous environments demands specialised techniques for sensor coupling, data acquisition and inversion routines." I disagree as it ignores arctic conditions where course materials also do exist.

*Modify accordingly within the text: "even though their employment on irregular rock surfaces and frozen environments demands specialized techniques for sensor coupling and data acquisition."*

- There is no need to say "ice (which is assumed to be an electrical insulator)" (line 144). Please be more concise and just say: "low conductivity". There is no need to artificially add more wording. This is a scientific publication, which should be factual and to the point.

*Modified accordingly within the text.*

- Finally, Section 5.5 seems to be out of context, i.e. not related to the actual work. The research is about the characterization of the two rock glaciers and not the assessment of the hydrological contribution. The authors have not presented any assessment or calculations that would support Section 5.5. For example, on line 331 the authors write: "Each have a distinct and important hydrological role." Based on the findings from the paper, one can argue that this might be true, and I understand that the authors follow this statement with some wording and thoughts. However, these statements do not follow a scientific line of arguments and assessments and should only be formulated in form of a working hypothesis that requires additional investigations or assessments.

*We now reformulate this section:*

*"El Ternero and El Jote represent two end-members of rock glacier types. El Ternero is an intact and likely active rock glacier containing significant amounts of ground ice according to our geophysical analysis, while El Jote is likely a relict rock glacier whose ice has largely if not completely melted. Each has a distinct and potentially important hydrological role. El Ternero has the capacity to function as long-term water storage given that most of the water it contains is in the form of ice which is insulated from the environment by debris cover (~5 m thick). The insulating effect of the debris cover has been shown to slow the rate of melt (Jones et al., 2018; Bonnaventure and Lamoureux, 2013) making rock glaciers more resilient to climate change compared to debris-free glaciers.*

*The petrophysical inversion model of water content suggests that El Jote contains water saturated bodies at depth, especially near the frontal slope. This interpretation is supported by the individual inversion model results including for media outside the rock glacier which is generally more conductive (1500Ωm) and interpreted as evidence of electrically resistive water presence in resistive material. The emergence of a perennial spring in a sloping peatland few hundred meters below points towards the existence of a proglacial aquifer, which may be connected to the rock glacier water saturated bodies. Thus, we infer that the relict rock glacier and the probable proglacial aquifer are storing and delaying the release of water downstream, assuming its hydrological role is similar to that of the relict rock glacier Schöneben in the Austrian Alps (Winkler et al., 2016). However, additional data are required to evaluate this hypothesis"*

*Moreover we address the specific comments on the manuscript in the attached pdf: tc-2020-306-referee-report#3*

[revised manuscript text omitted]

---

## Author Response (AR3)

The anonymous referee #1 pointed out different concerns about the paper that we try to address as follows:

- Hence, after long thinking, my suggestion is still that the authors provide a numerical study where they investigate the actual uncertainty in their results. Here they can provide 3 to 5 scenarios with varying ice/rock/air/water content and maybe lateral variations. I would recommend using exactly the same settings (quadrupoles and gather shots) as they use for the inversion of their results (i.e., after removal of outliers and geophones with poor coupling). Considering that the authors will have a real value (i.e., the numerical scenario), they can prove how good or bad their measuring setting can resolve different subsurface conditions, regarding both the geophysical heterogeneity and the air/water/ice/rock contents. The authors should also consider adding different noise levels.

  The numerical study could investigate the ability of the geometry used to collect the data to recover for instance the geometry of the rock glacier bottom (i.e., the contact to the bedrock) and the depth of the active layer. This is mentioned by the authors in line 27-28 as critical parts of the investigation of rock glaciers. Hence, the authors could establish a numerical study (with lateral variations in both the depth of the permafrost and the rock glacier) where ice content are defined as well defined anomalies and investigate how well the given geometry could recover such values. This may address the uncertainty in the ice and water content estimates. Maybe also to address components regarding the regularization that was mentioned during the very first version of the manuscript.

We now add a paragraph in the appendix (A2. results uncertainties) where we presents the results of a numerical simulation of the response of ERT and RST data to a known distribution of air/water/rock/ice for both field data settings and their petrophysical inversions. Here we present a quantification of the model results uncertainties accordingly to different noise levels in the simulated datasets.

In this new paragraph we address also the text-related specific comments of lines line 56-57, 73 and 284.

- The authors could also address the uncertainty associated to defining incorrect values for the Archie parameters (i.e., a, m n, and conductivity of the pore water).

As address in details later in the lines comment 225 and 241, these uncertainty and dependence on Archie parameters and conductivity of pore-water (in terms of validity of Archie's law depending on it) are addressed and discussed in the appendix (section A1 y A2). Also, in section A1 we discuss the regularization implemented in case of petrophysical joint inversion and we refer to Figure 3 and correspondent text for the regularization choice made in case of individual dataset inversion.

- Line 56 – 57 are incorrect and need to be addressed. The inversion is an ill-posed problem, not only affected by non-uniqueness. The presence of small uncertainty in the data generates large uncertainties in the inversion results. This is poorly addressed by the authors. The authors only discuss non-uniqueness. However, the influence of poor data quality has not been mentioned as a further source of error. Thus, I still believe that a numerical error could provide clear insights on the limitations of the results presented, especially the estimation of rock, air, water and ice content.

We modified the referred lines as follow:

*"Once the geophysical data have been collected, the information contained in these needs to be interpreted: geophysical inversion seeks to provide quantitative information about physical properties from indirect geophysical observations. This is an ill-posed problem whose solution is neither unique nor stable (Backus and Gilbert, 1970). Thus, if any set of model parameters can be found that is able to explain the observations, then an infinite number of parameters-sets would exist and arbitrarily small errors in the measurement data may lead to indefinitely large errors in the solutions (Kabanikhin 2008)"*

- Line 73: The joint inversion does not reduce the uncertainty. There is no computation of the uncertainty within the inversion algorithm. It helps to solve for a consistent model for the resistivity and the seismic data. Moreover, the solution converges to values that provide plausible estimations of rock, ice and air water content (as mentioned in the line 73 of the manuscript). However, there is no computation of uncertainties within the approach; this is still a deterministic solution. Hence, I believe that a numerical investigation based on their actual configuration (the measurements collected after removal of outliers) and simulating different conditions of the subsurface could really provide a estimation of the uncertainty in the results presented.

We follow the reviewer suggestions and re-write the sentence as:

*"joint inversion scheme aims solve a model that consistently explains both resistivity and seismic data sets and better constrains the components percentages in order to avoid nonphysical results"*

- Line 140 – 145 need to be carefully written. I provided some suggestions.

We follow the reviewer suggestions and re-write the sentence as:

*"In most rocks and soils, electrical current is carried by movements of ions in the pore water (electrolyte conduction) and by the movement of mobile ions in an electrical diffuse layer at the grain-fluid interface (surface conduction - Revil and Glover, 1997), with the mineral matrix generally characterized by high resistivity, unless electrical conductors are present within it…"*

- Line 151 – what is the contact resistivity? This needs to be corrected

Modified as *contact resistance*

- Line 175 – repeating and stacking sound redundant in the sentence. Maybe the authors could rephrase it?

We now deleted stacking

- Line 183 contains technical errors that need to be addressed

We modify the sentence as following:

*"…the dipole lengths for the potential measurements were of 5, 25 and 45 m, while for the current injections where between 15 and 235 m with intervals of 10 m…"*

- Line 203: Please provide clear description of the methods used to estimate the error parameters? Also about the parameters used to detect and remove outliers?

We do explain how the error parameters are estimate for both methods at lines 204-207 of the previously submitted manuscript:

*"…the error models resulted in 1.2% relative error for El Jote and 15% error for El Ternero, in the first case, the error was obtained from the average of the standard deviation for measured apparent resistivities whereas in the second case such average resulted in 11.4% but it was subsequently inflated to obtain a satisfactory inversion convergence. For the RST, an absolute error of 0.001 seconds was considered, estimated from the average variability of the first arrival picking. "*

Whereas we do explain the data processing in *section 3*.3 Data processing and Inversion:

*"The ERT observation were automatically filtered using the acquisition software for a standard deviation larger than 25%, while for the seismic refraction travel time we manually picked the first arrivals after applying a gain to the seismic traces, therefore the traces were filtered according to our ability of identify the first arrival times."*

- Line 225: which petrophysical constants values were used in the inversion of the data (i.e., a, m n, and conductivity of the pore water)? – was there any analysis on the dependence of the proposed results to those values?

We refer to section *A1 Inversion parameters for El Jote and EL Ternero* in the appendix, where we discuss the dependence of the inversion results on the Archie´s parameters *m* and *n* and on the initial porosity. As for the tortuosity factor *a* we followed the formulation fro Wagner et. Al (2019) and implement a modification of Archie's second law within the four phase petrophysical joint inversion (equation A3) where this parameter doesn't appears. We kindly refer to the comment regarding line 241 for the effects of a change in the resistivity model implemented within the petrophysical joint inversion.

- Line 241: if there are effects due to surface conductivity, could the authors discuss the applicability of the petrophysical joint inversion? – in the joint inversion deployed, the underlying model is only related to electrolytic conduction through Archie's model; thus, neglecting surface conductivity. If surface conductivity is dominating at the site, the estimates of rock, ice, water and air are incorrect . For instances, high surface conductivity are then interpreted as high porosity and water content. Yet this could be also related to lower porosity (i.e., due to the presence of fine grains), or higher ice content and the polarization of the EDL at the ice-water contact as mentioned in the line 242. This is an interesting point that the authors need to develop. However, it is only mentioned here to interpret the results and not addressed in the manuscript (i.e., lines 243 – 253).

We refer to section *A2 Methodology limitations in the appendix* (section A3 in the new submitted manuscript), where we describe the results of the analysis done by Mollaret et al. (2020). As explained in this section, they show that by using Archie's law, Archie's law with surface conduction and surface conduction model for the electrical resistivity petrophysical model, the inversion results are largely comparable and mainly depend on the porosity estimation, although they are based on theoretically different electrical conduction processes (due to the lack of field calibration of the respective electrical material parameters included in the equations, so that theses parameters are similarly determined by minimizing the data misfit).

- Line 284: what the authors imply with the marked sentence in the PDF? The P-wave velocity is defined by the ray paths within the inversion, not by other data. Maybe the authors mean that there are poorly resolved areas? – in such case, this needs to be rephrased. Is the poor coverage due to poor contact geophone-ground or due to the filtering within the processing of the data? – Maybe the numerical investigation can address for the particular measuring setting in this study (after removal of noisy measurements) and the ability of the data to detect anomalies within the poorly covered area are resolved?

We rephrase the sentence as follow:

*"the model parameter domains shown in the individual P-wave velocity inversion results and in the petrophysical joint inversion results (Figs. 4c, 6c, 5 and 7) are geometrically delimited by the lowermost ray path but in the P-wave velocity model presented there are poorly resolved areas due to the limited ray-coverage within the displayed area."*

- Line 347: I think the authors start an interesting topic by mentioning that "in many cases the implementation of petrophysical joint inversion can be limited by lack of proper petrophysical models". Could the authors develop further in this? - Do the authors consider that the 4-phase model implemented is not adequate? What petrophysical model needs to be implemented?

The quoted line is a general statement and doesn´t refer to the specific petrophysical model used in the study. Nevertheless we discuss its limitations in section A2 (section A3 in the new submitted manuscript) of the appendix

- Line 349-350: in which cases the proposed petrophysical inversion is not working? The authors should provide clear information. Address this directly with their data.

Please refer to comment above.

- Figure 6: the raypaths in the RST data could be plotted in a better y-axis to permit a better visualization of the data.

Modified accordingly

**Anonymous REFEREE #2**

The anonymous referee #2 pointed out different concerns about the paper that we try to address as follows:

- Title: As the study only presents data from one relict and one intact rock glacier, the title should be adjusted to: "Contrasting geophysical signature of a relict and an intact Andean rock glacier"

We modify the title accordingly.

- Line 1: Rock glaciers are not special types of glaciers as such it is scientifically wrong to say "other glacier types"

We modify the sentence as follow:

*"In semi-arid Chile, rock glaciers cover more surface area than glaciers and are potentially important water reserves"*

- Line 19: Herrington et al. (2018) did not write that the flow is coming from the rock glacier as the current wording implies. They have shown that water is passing through the rock glacier, which had an impact on water temperature and in consequence on the aquatic life. Please have a careful read at what Herrington et al. (2018) concluded.

We modified the sentence as following:

*"Harrington et al. (2018) investigated the hydrogeological characteristics of an inactive rock glacier in the Canadian Rockies, showing that the coarse blocky sediments forming the rock glacier allow the rapid infiltration of snowmelt and rain water to an unconfined aquifer above the bedrock surface. The water flowing through the aquifer is eventually routed via an internal channel parallel to the front of the rock glacier to a spring, which contributes up to 50% of basin streamflow during summer baseflow periods and up to 100% of basin streamflow over winter."*

- Line 32: What do you mean by "often unfrozen pore water"?

Water in liquid form, we now simplify this as: *"water"*

- Line 33: Cogley et al., 2011 is a really bad reference for that, as they only say: "Rock glacier: A mass of rock fragments and finer material in a matrix of ice, showing evidence of past or present flow."

We now removed this reference from the list and add *Jones, Darren B., et al. "Rock glaciers and mountain hydrology: A review." Earth-Science Reviews 193 (2019): 66-90.*

- Line 40: I encourage the authors to cite the IPA Rock Glacier Action Group documents: https://www.unifr.ch/geo/geomorphology/en/research/ipa-action-group-rock-glacier/

We now add the suggested reference

- Line 90: why is this unpublished? The inventory is publicly available.

This inventory has been published. We have now included the web address where this inventory can be downloaded in the text (Available from: https://dga.mop.gob.cl/estudiospublicaciones/mapoteca/Paginas/Mapoteca-Digital.aspx)

- Line 95: I suggest deleting "mining" since the mining industry actually doesn't need as much water as other industries. I therefore suggest to simply combining all industries and state "industry".

Nevertheless it is the main industry present in the region (and with the agriculture sector the main economical activity too) therefore we would like to leave the specification

- Line 148: The inclusion to Table 1 isn't needed. Those are values cited many times in other references and as such a cross reference will just do.

We now modify the sentence as follow and delete Table 1:

*"Relevant values for electrical resistivity in rock glacier environments may be found in Maurer and Hauck (2007) and Hauck and Kneisel (2008)"*

- Line 164: Delete the reference to Table 1

Modified accordingly.

- Line 246: Make sure you use sediments and not rocks: "unconsolidated sediments". This not only applies in this line, i.e., check the whole manuscript.

As commented in a previous round of revision, we would like to keep the terms rock for coherence with the original four phase model formulation afterwards used for the petrophysical joint inversion.

- Line 249: It should be "debris", not "debrids". This typo also appears at other places, so please check the manuscript.

Modified accordingly

- Line 269: Based on the electrode spacing I assume that error in the estimation of the active layer thickness is at least 25%. Is that correct, and if so, please add the error to the estimate.

We now add this information within the text.

- Line 341: "Dos Lenguas rock glacier"

Modified accordingly

- Line 342: "volumetric ice content"

Modified accordingly

- Line 342: Delete the % sign from the text. A unit sign (keep in mind % is a unit) only appears behind a number. If you want to use it in the text, write "percentage". Check the whole manuscript as you can probably delete the symbol in most places anyway.

Modified accordingly along the manuscript.

- Line 373: This Section 5.5 does not provide value and should be deleted. The authors should focus on the objective of the manuscript, which is the geophysical investigation. The discussion on hydrogeology, in fact it is a discussion on hydrology only, is completely insufficient. Many key references are missing, and such a discussion would need to be expanded significantly. If this is an important topic for the authors, I encourage them to partner with a hydrologist and a Hydrogeologists and write an independent paper that is dedicated to that topic. In its current form it is inadequate and should not be accepted. It's telling that this aspect isn't reflected in the conclusion or outlook section. Not even the key sentence from Section 5.5 (Line 387-388).

We now deleted section 5.5 from the paper

- Line 406: replace "complicated" with "challenging".

Modified accordingly

- Figure 1: Elevations in masl.

Modified accordingly also in figure 2

- Figure 9: Middle image should be b).

Modified accordingly